# Gradient-Free Methods for Nonconvex Nonsmooth Stochastic Compositional Optimization

**Zhuanghua Liu**
Department of Computer Science, National University of Singapore
CNRS@CREATE LTD, 1 Create Way, #08-01 CREATE Tower, Singapore 138602
liuzhuanghua9@gmail.com

**Luo Luo**[*]
School of Data Science, Fudan University
Shanghai Key Laboratory for Contemporary Applied Mathematics
luoluo@fudan.edu.cn

**Bryan Kian Hsiang Low**
Department of Computer Science, National University of Singapore
lowkh@comp.nus.edu.sg

## Abstract

Stochastic compositional optimization (SCO) problems are popular in many real-world applications, including risk management, reinforcement learning, and meta-learning. However, most of the previous methods for SCO require the smoothness assumption on both the outer and inner functions, which limits their applications to a wider range of problems. In this paper, we study the SCO problem in that both the outer and inner functions are Lipschitz continuous but possibly nonconvex and nonsmooth. In particular, we propose gradient-free stochastic methods for finding the $(\delta, \epsilon)$-Goldstein stationary points of such problems with non-asymptotic convergence rates. Our results also lead to an improved convergence rate for the convex nonsmooth SCO problem. Furthermore, we conduct numerical experiments to demonstrate the effectiveness of the proposed methods.

## 1 Introduction

In this paper, we consider the following stochastic compositional optimization (SCO) problem:

$$\min_{\mathbf{x} \in \mathbb{R}^d} \Phi(\mathbf{x}) \triangleq f(g(\mathbf{x})), \tag{1}$$

where the outer and inner functions $f \colon \mathbb{R}^m \to \mathbb{R}$ and $g \colon \mathbb{R}^d \to \mathbb{R}^m$ has the form of

$$f(\mathbf{y}) \triangleq \mathbb{E}_{\boldsymbol{\xi}}[F(\mathbf{y}; \boldsymbol{\xi})] \qquad \text{and} \qquad g(\mathbf{x}) \coloneqq \mathbb{E}_{\boldsymbol{\zeta}}[G(\mathbf{x}; \boldsymbol{\zeta})],$$

and the stochastic components $F(\mathbf{y}; \boldsymbol{\xi})$ and $G(\mathbf{x}; \boldsymbol{\zeta})$ are Lipschitz continuous but possibly nonconvex and nonsmooth. Random variables $\boldsymbol{\xi}$ and $\boldsymbol{\zeta}$ are independent. Such formulation is popular in many real-world applications, including risk management [1], statistical learning [2], reinforcement learning [3], and model agnostic meta-learning [4].

Most of the existing work [2, 5, 6, 7, 8] for nonconvex SCO problem is based on the assumption that both functions $f(\cdot)$ and $g(\cdot)$ are smooth. Unfortunately, many modern machine learning

---

[*]The corresponding author

38th Conference on Neural Information Processing Systems (NeurIPS 2024).

Table 1: We present the stochastic zeroth-order complexity of proposed algorithms for solving nonsmooth stochastic compositional optimization problems.

| METHODS | PROBLEM | COMPLEXITY | REFERENCE |
|---------|---------|------------|-----------|
| GFCOM | NONCONVEX | $\mathcal{O}(d^{3.5}\delta^{-3}\epsilon^{-6})$ | COROLLARY 4.2 |
| GFCOM$^+$ | NONCONVEX | $\mathcal{O}(d^{3.5}\delta^{-3}\epsilon^{-5})$ | COROLLARY 4.4 |
| WS-GFCOM$^2$ | CONVEX | $\mathcal{O}(d^{3}\delta^{-2.4}\epsilon^{-4.8} + d^{3.5}\delta^{-2}\epsilon^{-6})$ | COROLLARY 5.3 |
| WS-GFCOM$^+$ | CONVEX | $\mathcal{O}(d^{3}\delta^{-2.4}\epsilon^{-4} + d^{3.5}\delta^{-2}\epsilon^{-5})$ | COROLLARY 5.4 |

models including deep neural networks do not satisfy the smoothness condition. Ruszczynski [9] proposed a single time-scale stochastic subgradient method for solving the Problem (1). However, the author only provided asymptotic convergence analysis for the approach. In recent work, Liu and Davanloo Tajbakhsh [10], Hu et al. [11] presented the non-asymptotic convergence for the nonconvex nonsmooth SCO problem, while their analysis requires additional assumptions such as the weak-convexity and the relative smoothness condition.

The non-smoothness in the Problem (1) implies the classical gradient-based approaches and the convergence measure in terms of the gradient norm cannot be applied. The Clarke subdifferential [12] for the Lipshitz continuous functions is a natural extension of gradients for the smooth functions. However, hard instances have shown that no deterministic or randomized algorithms can find an $\epsilon$-stationary point with respect to the Clarke subdifferential of a Lipschitz function in finite time [13, 14]. To address this issue, Zhang et al. [13] proposed a refined notion of the $(\delta, \epsilon)$-Goldstein stationary point in terms of the Goldstein $\delta$-subdifferential, which considers the convex hull of the Clarke subdifferential at points in the $\delta$-neighbourhood [15].

In this paper, we propose a zeroth-order stochastic method called gradient-free compositional optimization method (GFCOM) for solving Problem (1) in finite time. In particular, we show that the GFCOM can find a $(\delta, \epsilon)$-Goldstein stationary point of the objective function $\Phi(\cdot) = f(g(\cdot))$ within the stochastic zeroth-order oracle complexity of $\mathcal{O}(d^{3.5}\delta^{-3}\epsilon^{-6})$. Furthermore, we improve the GFCOM by using the variance reduction technique [16, 17, 18, 19] to establish a more efficient first-order oracle estimator, leading to the algorithm GFCOM$^+$ which achieves a tighter upper complexity bound of $\mathcal{O}(d^{3.5}\delta^{-3}\epsilon^{-5})$. In addition, we study convex nonsmooth SCO problems. In this regime, prior methods [2, 20, 21] suffer two major limitations: (i) Their convergence analysis is based on the smoothness condition of the outer function. (ii) Their convergence result is measured by the sub-optimality of the function value gap, while the non-asymptotic convergence rate for finding the stationary point has not been studied. We overcome these issues by involving a warm-start strategy into GFCOM$^+$, which is guarantee to find a $(\delta, \epsilon)$-Goldstein stationary point within $\mathcal{O}(d^{3}\delta^{-2.4}\epsilon^{-4} + d^{3.5}\delta^{-2}\epsilon^{-5})$ stochastic zeroth-order oracle complexity. We summarize the complexity of proposed methods in Table 1.

## 2 Related Work

In this section, we review prior work for stochastic compositional optimization and classical nonconvex nonsmooth optimization.

### 2.1 Stochastic Compositional Optimization

In a pioneer work, Wang et al. [2] studied the non-asymptotic convergence of nonconvex smooth stochastic compositional optimization by proposing the stochastic compositional gradient descent (SCGD), which contains two sequences of stepsizes for different time scales to update the variable and track the inner function value, respectively. The authors also heuristically extended their methods to zeroth-order optimization. Wang et al. [5] proposed an accelerated variant of SCGD using an extrapolation-smoothing scheme, and Ghadimi et al. [6] proposed a single time-scale approach to accelerate the convergence further. Additionally, Hu et al. [7], Lin et al. [20], Yuan et al. [22] incorporated the variance reduction technique into the first-order iteration, achieving a tight stochastic first-order complexity under the mean-squared smoothness assumption.

Compared with the smooth counterpart, the study of nonsmooth compositional optimization is relatively scarce. Ruszczynski [9] proposed a single time-scale stochastic subgradient method for nonconvex nonsmooth SCO problems, while the theoretical analysis only provided the asymptotic convergence rate. Liu and Davanloo Tajbakhsh [10] introduced the stochastic composition Bregman gradient method and provided a non-asymptotic convergence analysis under the relative smoothness condition. Vladarean et al. [23] proposed a Frank-Wolfe algorithm for constrained nonconvex nonsmooth SCO problems. Their analysis assumes that the outer function is convex but possibly non-differentiable, and the inner function is smooth. Very recently, Hu et al. [11] studied stochastic methods for the finite-sum coupled compositional optimization problem. Their convergence rate is established by assuming both the outer and inner functions are weakly convex, and the outer function is non-decreasing. In addition, Kalogerias and Powell [24] studied the zeroth-order stochastic optimization for a specific compositional optimization problem in risk-aware learning.

## 2.2 Non-Asymptotic convergence Analysis of Nonconvex Nonsmooth Optimization

In this subsection, we present a literature review for classical nonconvex nonsmooth optimization. The study of this field has a long history [12, 25], but the non-asymptotic convergence analysis of nonsmooth optimization has only emerged in recent years. Zhang et al. [13] provided the non-asymptotic complexity analysis of the interpolated normalized gradient descent method to achieve a $(\delta, \epsilon)$-Goldstein stationary point of a Lipschitz function with a nonstandard subgradient oracle. Davis et al. [26], Tian et al. [27] improved this method by introducing random perturbations in each iteration to remove the assumptions. Recently, Cutkosky et al. [28] proposed the optimal algorithm via the reduction from nonconvex nonsmooth optimization to online learning.

The development of non-asymptotic convergence analysis of zeroth-order methods for nonsmooth optimization was initiated by Nesterov and Spokoiny [29]. Later, Lin et al. [30] proposed gradient-free methods for this problem by establishing a relationship between the Goldstein $\delta$-subdifferential and randomized smoothing. Chen et al. [31], Liu et al. [32] improved their results by leveraging the variance-reduction technique. Kornowski and Shamir [33] obtained a sharper bound by applying the reduction technique introduced by Cutkosky et al. [28] to the gradient-free setting. Liu et al. [34], Grimmer and Jia [35] further extends the methodology to the constrained setting. However, these methods do not apply to the nonconvex nonsmooth SCO Problem (1).

# 3 Preliminaries

In this section, we first present the notations and assumptions used in this paper, then introduce the convergence criteria for nonsmooth optimization and the randomized smoothing technique.

## 3.1 Notations and Assumptions

We use $\|\cdot\|$ to denote the Euclidean norm of a vector. We define $\mathbb{B}_\delta(\mathbf{x}) \triangleq \{\mathbf{y} \in \mathbb{R}^d : \|\mathbf{y} - \mathbf{x}\| \leq \delta\}$ as the Euclidean ball centered at $\mathbf{x} \in \mathbb{R}^d$ with a radius $\delta > 0$. We let $\mathrm{conv}(A)$ be the convex hull of the set $A$. For two given sets $A$ and $B$, we define $A \times B$ as their Cartesian product. In addition, we denote $f \circ g$ as the function composition such that $(f \circ g)(\mathbf{x}) \triangleq f(g(\mathbf{x}))$.

Throughout this paper, we assume the objective function (1) satisfies the following assumptions.

**Assumption 3.1.** *We assume the stochastic component $F(\cdot, \boldsymbol{\xi})$ is $L_f(\boldsymbol{\xi})$-Lipschitz for any given $\boldsymbol{\xi}$, and the stochastic component $G(\cdot, \boldsymbol{\zeta})$ is $L_g(\boldsymbol{\zeta})$-Lipschitz for any given $\boldsymbol{\zeta}$. That is, it holds*

$$|F(\mathbf{x}, \boldsymbol{\xi}) - F(\mathbf{y}, \boldsymbol{\xi})| \leq L_f(\boldsymbol{\xi}) \|\mathbf{x} - \mathbf{y}\| \quad and \quad \|G(\hat{\mathbf{x}}, \boldsymbol{\zeta}) - G(\hat{\mathbf{y}}, \boldsymbol{\zeta})\| \leq L_g(\boldsymbol{\zeta}) \|\hat{\mathbf{x}} - \hat{\mathbf{y}}\|,$$

*for any $\mathbf{x}, \mathbf{y} \in \mathbb{R}^d$ and $\hat{\mathbf{x}}, \hat{\mathbf{y}} \in \mathbb{R}^m$. We also assume the Lipschitz parameters $L_f(\boldsymbol{\xi})$ and $L_g(\boldsymbol{\zeta})$ have bounded second-order moments such that $\mathbb{E}_{\boldsymbol{\xi}}[L_f(\boldsymbol{\xi})^2] \leq G_f^2$ and $\mathbb{E}_{\boldsymbol{\zeta}}[L_g(\boldsymbol{\zeta})^2] \leq G_g^2$ for some constants $G_f, G_g > 0$.*

*Remark* 3.2. We can verify that Assumption 3.1 implies the function $f(\cdot)$ is $G_f$-Lipschitz, and the function $g(\cdot)$ is $G_g$-Lipschitz by Jensen's inequality.

**Assumption 3.3.** *We assume that there exists a constant $\sigma_0$ as the upper bound on the variance of the functions $G(\cdot, \boldsymbol{\zeta})$, such that for any $\mathbf{x} \in \mathbb{R}^d$, we have $\mathbb{E}_{\boldsymbol{\zeta}} \left[ \|G(\mathbf{x}, \boldsymbol{\zeta}) - g(\mathbf{x})\|^2 \right] \leq \sigma_0^2$.*

**Assumption 3.4.** *We assume that the composite function $\Phi(\cdot) \triangleq (f \circ g)(\cdot)$ is lower bounded such that $\Phi^* \triangleq \inf_{\mathbf{x} \in \mathbb{R}^d} \Phi(\mathbf{x}) > -\infty$.*

## 3.2 Convergence Criteria for Nonsmooth Functions

We introduce the definitions of the Clarke subdifferential and approximate Clarke stationary points.

**Definition 3.5** (Clarke [12])**.** *The Clarke subdifferential of a Lipschitz function $f$ at a point $\mathbf{x}$ is defined as $\partial f(\mathbf{x}) \triangleq \operatorname{conv}\{g : g = \lim_{\mathbf{x}_k \to \mathbf{x}} \nabla f(\mathbf{x}_k)\}$. Furthermore, we call a point $\mathbf{x}$ an $\epsilon$-Clarke stationary point of $f$ if it holds that $\min\{\|g\| : g \in \partial f(\mathbf{x})\} \leq \epsilon$.*

Zhang et al. [13], Kornowski and Shamir [14] showed that no deterministic or randomized algorithm could find an $\epsilon$-Clarke stationary point in finite time. Consequently, Zhang et al. [13] considered a refined notion of approximate stationary point in terms of the Goldstein $\delta$-subdifferential.

**Definition 3.6** (Zhang et al. [13])**.** *Given a Lipschitz function $f : \mathbb{R}^d \to \mathbb{R}$ and $\delta > 0$, the Goldstein $\delta$-subdifferential of $f$ at point $\mathbf{x} \in \mathbb{R}^d$ is defined as $\partial_\delta f(\mathbf{x}) := \operatorname{conv}(\cup_{\mathbf{y} \in \mathbb{B}_\delta(\mathbf{x})} \partial f(\mathbf{y}))$, which is the convex hull of the Clarke subdifferential at the points in the $\delta$-neighbourhood of $\mathbf{x}$. Additionally, a point $\mathbf{x} \in \mathbb{R}^d$ is called a $(\delta, \epsilon)$-Goldstein stationary point of $f(\cdot)$ if it holds that $\min\{\|g\| : g \in \partial_\delta f(\mathbf{x})\} \leq \epsilon$.*

Recent work [13, 26, 28] has shown that it is possible to find a $(\delta, \epsilon)$-Goldstein stationary point of a nonsmooth problem without a composition structure in finite time. However, these theories are not applicable to nonsmooth SCO. In particular, we can infer from Rademacher's theorem and Assumption 3.1 that the composite function $\Phi(\cdot)$ is differentiable almost everywhere. Let $\mathcal{Q} \subseteq \mathbb{R}^d$ be the set on which $\Phi$ is differentiable, then $\mathbb{R}^d \setminus \mathcal{Q}$ is of measure zero. Recent work assumes they have access to the unbiased stochastic gradient estimator of the objective function for any $\mathbf{x} \in \mathcal{Q}$. In our setting, the unbiased gradient estimator of the composite function $\Phi(\mathbf{x})$ is $\mathcal{J}_G(\mathbf{x}; \boldsymbol{\zeta}) \nabla F(g(\mathbf{x}); \boldsymbol{\xi})$, where $\mathcal{J}_G$ is the Jacobian matrix of the function $G(\cdot; \boldsymbol{\zeta})$. However, such an estimator is hard to obtain because the function value $g(\mathbf{x})$ is an expectation.

## 3.3 Randomized Smoothing

The randomized smoothing is a popular technique for nonsmooth analysis [36] and gradient-free optimization [29]. Concretely, given a Lipschitz function $f$ and a uniform distribution $\mathcal{P}$ on a unit ball, we define its smoothed surrogate as $f_\delta(\mathbf{x}) = \mathbb{E}_{\mathbf{u} \sim \mathcal{P}}[f(\mathbf{x} + \delta \mathbf{u})]$, which has the following properties.

**Lemma 3.7** (Lin et al. [30, Proposition 2.3])**.** *Let $f_\delta(\mathbf{x}) = \mathbb{E}_{\mathbf{u} \sim \mathcal{P}}[f(\mathbf{x} + \delta \mathbf{u})]$ where $\mathcal{P}$ is a uniform distribution on a unit ball in $\ell_2$-norm. Suppose the function $f$ is $L$-Lipschitz, then we have (a) $|f_\delta(\mathbf{x}) - f(\mathbf{x})| \leq \delta L$; (b) $f_\delta(\cdot)$ is differentiable everywhere and $L$-Lipschitz with $cL\sqrt{d}\delta^{-1}$-Lipschitz gradient, where $c$ is some positive constant; (c) $\nabla f_\delta(\mathbf{x}) \in \partial_\delta f(\mathbf{x})$ for any $\mathbf{x} \in \mathbb{R}^d$.*

Moreover, we consider the following unbiased gradient estimator of the smoothed surrogate function $f_\delta(\cdot)$, which can be obtained from two function query oracle calls on points uniformly sampled from a unit sphere [37].

**Lemma 3.8** (Lin et al. [30, Lemma D.1])**.** *Let $f(\mathbf{x}) = \mathbb{E}[F(\mathbf{x}; \boldsymbol{\xi})]$ be a $L$-Lipschitz function. We denote*

$$\iota(\mathbf{x}; \mathbf{u}, \boldsymbol{\xi}) \triangleq \frac{d}{2\delta}(F(\mathbf{x} + \delta \mathbf{u}; \boldsymbol{\xi}) - F(\mathbf{x} - \delta \mathbf{u}; \boldsymbol{\xi}))\mathbf{u},$$

*where $\mathbf{u}$ is uniformly sampled from a distribution on a unit sphere in $\mathbb{R}^d$ space. Then, we have $\mathbb{E}[\iota(\mathbf{x}; \mathbf{u}, \boldsymbol{\xi})] = \nabla f_\delta(\mathbf{x})$ and $\mathbb{E}[\|\iota(\mathbf{x}; \mathbf{u}, \boldsymbol{\xi})\|^2] \leq 16\sqrt{2\pi}dL^2$.*

# 4 Algorithms for Nonconvex Nonsmooth SCO

In this section, we propose zeroth-order stochastic algorithms for solving the nonconvex nonsmooth SCO problem. We also provide non-asymptotic convergence analysis for the proposed methods.

**Algorithm 1:** GFCOM($\mathbf{x}_0, \eta, T, b_f, b_g$)

---

**1 for** $t = 0, 1, \ldots, T - 1$ **do**

2      Sample $\{\boldsymbol{\xi}_{t,i}, \mathbf{w}_{t,i}\}_{i=1}^{b_f}$ and $\{\boldsymbol{\zeta}_{t,i}\}_{i=1}^{b_g}$.

3      Generate $G(\mathbf{x}_t \pm \delta\mathbf{w}_{t,j}; \boldsymbol{\zeta}_{t,i})$ for every $(i,j) \in [b_g] \times [b_f]$.

4      Let $\mathbf{y}_{t,j} = \frac{1}{b_g} \sum_{i \in [b_g]} G(\mathbf{x}_t + \delta\mathbf{w}_{t,j}; \boldsymbol{\zeta}_{t,i})$.

5      Let $\mathbf{z}_{t,j} = \frac{1}{b_g} \sum_{i \in [b_g]} G(\mathbf{x}_t - \delta\mathbf{w}_{t,j}; \boldsymbol{\zeta}_{t,i})$.

6      Let $\mathbf{v}_t = \frac{1}{b_f} \sum_{j \in [b_f]} \frac{d}{2\delta} (F(\mathbf{y}_{t,j}; \boldsymbol{\xi}_{t,j}) - F(\mathbf{z}_{t,j}; \boldsymbol{\xi}_{t,j})) \mathbf{w}_{t,j}$.

7      Update $\mathbf{x}_{t+1} = \mathbf{x}_t - \eta\mathbf{v}_t$.

**8 end**

**9 Return:** $\mathbf{x}_R$ where $R$ is uniformly sampled from $[T]$.

---

**Algorithm 2:** GFCOM$^+$($\mathbf{x}_0, \eta, T, b_f, b_f', b_g, b_g', m$)

---

**1 for** $t = 0, 1, \ldots, T - 1$ **do**

2      **if** $t \bmod m = 0$ **then**

3          Sample $\{\boldsymbol{\xi}_{t,i}, \mathbf{w}_{t,i}\}_{i=1}^{b_f}$ and $\{\boldsymbol{\zeta}_{t,i}\}_{i=1}^{b_g}$.

4          Generate $G(\mathbf{x}_t \pm \delta\mathbf{w}_{t,j}; \boldsymbol{\zeta}_{t,i})$ for every $(i,j) \in [b_g] \times [b_f]$.

5          Let $\mathbf{y}_{t,j} = \frac{1}{b_g} \sum_{i \in [b_g]} G(\mathbf{x}_t + \delta\mathbf{w}_{t,j}; \boldsymbol{\zeta}_{t,i})$.

6          Let $\mathbf{z}_{t,j} = \frac{1}{b_g} \sum_{i \in [b_g]} G(\mathbf{x}_t - \delta\mathbf{w}_{t,j}; \boldsymbol{\zeta}_{t,i})$.

7          Let $\mathbf{v}_t = \frac{1}{b_f} \sum_{j \in [b_f]} \frac{d}{2\delta} (F(\mathbf{y}_{t,j}; \boldsymbol{\xi}_{t,j}) - F(\mathbf{z}_{t,j}; \boldsymbol{\xi}_{t,j})) \mathbf{w}_{t,j}$.

8      **else**

9          Sample $\{\boldsymbol{\xi}_{t,i}, \mathbf{w}_{t,i}\}_{i=1}^{b_f'}$ and $\{\boldsymbol{\zeta}_{t,i}\}_{i=1}^{b_g'}$.

10        Generate $G(\mathbf{x}_t \pm \delta\mathbf{w}_{t,j}; \boldsymbol{\zeta}_{t,i})$ and $G(\mathbf{x}_{t-1} \pm \delta\mathbf{w}_{t,j}; \boldsymbol{\zeta}_{t,i})$ for every $(i,j) \in [b_g'] \times [b_f']$.

11        Let $\mathbf{y}_{k,j} = \frac{1}{b_g'} \sum_{i \in [b_g']} G(\mathbf{x}_k + \delta\mathbf{w}_{t,j}; \boldsymbol{\zeta}_{k,i})$ for $k \in \{t-1, t\}$.

12        Let $\mathbf{z}_{k,j} = \frac{1}{b_g'} \sum_{i \in [b_g']} G(\mathbf{x}_k - \delta\mathbf{w}_{t,j}; \boldsymbol{\zeta}_{k,i})$ for $k \in \{t-1, t\}$.

13        Let $\mathbf{q}_k = \frac{1}{b_f'} \sum_{j \in [b_f']} \frac{d}{2\delta} (F(\mathbf{y}_{k,j}; \boldsymbol{\xi}_{t,j}) - F(\mathbf{z}_{k,j}; \boldsymbol{\xi}_{t,j})) \mathbf{w}_{t,j}$ for $k \in \{t-1, t\}$.

14        Let $\mathbf{v}_t = \mathbf{q}_t - \mathbf{q}_{t-1} + \mathbf{v}_{t-1}$.

**15**      **end**

16      Update $\mathbf{x}_{t+1} = \mathbf{x}_t - \eta\mathbf{v}_t$.

**17 end**

**18 Return:** $\mathbf{x}_R$ where $R$ is uniformly sampled from $[T]$.

---

### 4.1 The Algorithms

In this subsection, we propose the gradient-free compositional optimization method (GFCOM) and its accelerated variant GFCOM$^+$. We first introduce the main intuition of the GFCOM. Consider the following hypothetical zeroth-order gradient estimator

$$\bar{\mathbf{v}}_t = \frac{1}{b_f} \sum_{j \in [b_f]} \frac{d}{2\delta} (F(g(\mathbf{x}_t + \delta\mathbf{w}_{t,j}); \boldsymbol{\xi}_{t,j}) - F(g(\mathbf{x}_t - \delta\mathbf{w}_{t,j}); \boldsymbol{\xi}_{t,j})) \mathbf{w}_{t,j}, \tag{2}$$

where $b_f > 0$ is the mini-batch size of the gradient estimator. By Lemma 3.8, the vector $\bar{\mathbf{v}}_t$ is an unbiased estimator of $\nabla\Phi_\delta(\mathbf{x}_t)$. Unfortunately, it is intractable to obtain the function values $g(\mathbf{x}_t \pm \delta\mathbf{w}_{t,j})$ because $g(\cdot)$ is an expectation of stochastic component functions $G(\cdot; \boldsymbol{\zeta})$. To remedy this issue, we introduce auxiliary variables $\mathbf{y}_{t,j}$ and $\mathbf{z}_{t,j}$ to approximate the inner function values $g(\mathbf{x}_t + \delta\mathbf{w}_{t,j})$ and $g(\mathbf{x}_t - \delta\mathbf{w}_{t,j})$, respectively. In particular, the vectors $\mathbf{y}_{t,j}$ and $\mathbf{z}_{t,j}$ are mini-batch function estimators defined as follows

$$\mathbf{y}_{t,j} = \frac{1}{b_g} \sum_{i \in [b_g]} G(\mathbf{x}_t + \delta\mathbf{w}_{t,j}; \boldsymbol{\zeta}_{t,i}), \quad \text{and} \quad \mathbf{z}_{t,j} = \frac{1}{b_g} \sum_{i \in [b_g]} G(\mathbf{x}_t - \delta\mathbf{w}_{t,j}; \boldsymbol{\zeta}_{t,i}), \tag{3}$$

where $b_g > 0$ is the mini-batch size of the function estimator. Accordingly, we use these two variables to replace the function calls of $g(\cdot)$ in the gradient estimator $\mathbf{v}_t$ of Eq. (2). The complete procedure of the GFCOM is presented in Algorithm 1.

For the GFCOM⁺, we leverage the variance reduction technique to approximate $\nabla\Phi_\delta(\mathbf{x}_t)$. In particular, we consider the following hypothetical recursive gradient estimator

$$\bar{\mathbf{v}}_t = \bar{\mathbf{q}}_t - \bar{\mathbf{q}}_{t-1} + \mathbf{v}_{t-1}, \tag{4}$$

where $\bar{\mathbf{q}}_t$ and $\bar{\mathbf{q}}_{t-1}$ are mini-batch gradient estimator defined as follows.

$$\bar{\mathbf{q}}_k = \frac{1}{b'_f} \sum_{j \in [b'_f]} \frac{d}{2\delta}(F(g(\mathbf{x}_k + \delta\mathbf{w}_{t,j}); \boldsymbol{\xi}_{t,j}) - F(g(\mathbf{x}_k - \delta\mathbf{w}_{t,j}); \boldsymbol{\xi}_{t,j}))\mathbf{w}_{t,j},$$

where $b'_f > 0$ is the mini-batch size and $k \in \{t-1, t\}$. Compared with the mini-batch gradient estimator (2), the recursive gradient estimator (4) has been shown to achieve a sharper complexity bound in nonconvex optimization literature [17, 18, 31]. However, the gradient estimator is computationally intractable due to the unknown function $g(\cdot)$. Similar to the development of Algorithm 1, we define $\mathbf{y}_t, \mathbf{z}_t$ to estimate the inner function values $g(\mathbf{x}_t \pm \delta\mathbf{w}_{t,j})$. We also introduce variables $\mathbf{y}_{t-1}, \mathbf{z}_{t-1}$ to approximate the inner function values $g(\mathbf{x}_{t-1} \pm \delta\mathbf{w}_{t,j})$ at the previous iteration. Then we define stochastic gradient estimators $\mathbf{q}_t$ and $\mathbf{q}_{t-1}$ in terms of $\mathbf{y}_t, \mathbf{y}_{t-1}, \mathbf{z}_t$ and $\mathbf{z}_{t-1}$.

$$\mathbf{q}_k = \frac{1}{b'_f} \sum_{j \in [b'_f]} \frac{d}{2\delta}(F(\mathbf{y}_{k,j}; \boldsymbol{\xi}_{t,j}) - F(\mathbf{z}_{k,j}; \boldsymbol{\xi}_{t,j}))\mathbf{w}_{t,j},$$

for $k \in \{t-1, t\}$. We replace the minibatch gradient estimator $\bar{\mathbf{q}}_t$ and $\bar{\mathbf{q}}_{t-1}$ in the recursive gradient estimator $\bar{\mathbf{v}}_t$ of Eq. (4) with the refined gradient estimators $\mathbf{q}_t$ and $\mathbf{q}_{t-1}$. The complete procedure of GFCOM⁺ is presented in Algorithm 2.

## 4.2 Convergence Analysis

In this subsection, we consider the complexity analysis of the proposed algorithms introduced in Section 4.1. We assume that $\Phi(x_0) - \Phi^* \leq R$, where $R > 0$ is some constant.

The following theorem shows the convergence rate of solving the Problem (1) with the GFCOM method presented in Algorithm 1.

**Theorem 4.1.** *Under Assumption 3.1, 3.3 and 3.4, running the GFCOM algorithm (Algorithm 1) with $\eta \leq \delta/(cG_f G_g \sqrt{d})$ where $c > 0$ is some constant, then the output $\mathbf{x}_R$ satisfies*

$$\mathbb{E}\left[\|\nabla\Phi_\delta(\mathbf{x}_R)\|^2\right] = \mathcal{O}\left(\frac{G_f G_g \sqrt{d}R}{\delta T} + \frac{G_f^2 G_g^2 \sqrt{d}}{T} + \frac{dG_f^2 G_g^2}{b_f} + \frac{d^2 G_f^2 \sigma_0^2}{\delta^2 b_g}\right). \tag{5}$$

Using Theorem 4.1 with the parameter setting

$$T = \Theta\left(\frac{G_f G_g \sqrt{d}R}{\delta\epsilon^2} + \frac{G_f^2 G_g^2 \sqrt{d}}{\epsilon^2}\right), \quad b_f = \Theta\left(\frac{dG_f^2 G_g^2}{\epsilon^2}\right) \quad \text{and} \quad b_g = \Theta\left(\frac{d^2 G_f^2 \sigma_0^2}{\delta^2 \epsilon^2}\right),$$

we obtain the following oracle complexity result for Algorithm 1.

**Corollary 4.2.** *Under Assumption 3.1, 3.3 and 3.4, the GFCOM algorithm (Algorithm 1) requires at most $\mathcal{O}\left(d^{3.5}G_f^5 G_g^3 \sigma_0^2 R\delta^{-3}\epsilon^{-6} + d^{3.5}G_f^6 G_g^4 \sigma_0^2 \delta^{-2}\epsilon^{-6}\right)$ stochastic zeroth-order function query calls to obtain a $(\delta, \epsilon)$-Goldstein stationary point of $\Phi$.*

After giving the complexity bound of GFCOM in Corollary 4.2, we now consider the convergence analysis of GFCOM⁺. We will show that it enjoys a sharper complexity bound due to the utilization of the recursive gradient estimator. The following theorem shows the convergence rate of solving Problem (1) with the GFCOM⁺ (Algorithm 2).

**Theorem 4.3.** *Under Assumption 3.1, 3.3 and 3.4, running the GFCOM⁺ algorithm (Algorithm 2) with $\eta = \delta/(2cG_f G_g \sqrt{d})$, $b'_f = \Theta(dG_f G_g \epsilon^{-1})$ and $m = \Theta(G_f G_g \epsilon^{-1})$, then the output $\mathbf{x}_R$ satisfies*

$$\mathbb{E}\left[\|\nabla\Phi_\delta(\mathbf{x}_R)\|^2\right] = \mathcal{O}\left(\frac{\sqrt{d}G_f G_g R}{\delta T} + \frac{\sqrt{d}G_f^2 G_g^2}{T} + \frac{dG_f^2 G_g^2}{b_f} + \frac{d^2 G_f^2 \sigma_0^2}{\delta^2 b_g} + \frac{d^2 G_f^2 \sigma_0^2}{\delta^2 b'_g}\right).$$

---

**Algorithm 3:** WS-GFCOM($\mathbf{x}_0, \eta_0, T_0, b_{g,0}, \eta, T, b_f, b_g, b'_f, b'_g, m$)

---

**1** Let $\mathbf{x}_1 = \text{GFCOM}(\mathbf{x}_0, \eta_0, T_0, 1, b_{g,0})$.

**2** **Option I (WS-GFCOM²):** $\mathbf{x}_2 = \text{GFCOM}(\mathbf{x}_1, \eta, T, b_f, b_g)$.

**3** **Option II (WS-GFCOM⁺):** $\mathbf{x}_2 = \text{GFCOM}^+(\mathbf{x}_1, \eta, T, b_f, b'_f, b_g, b'_g, m)$.

**4** **Return:** $\mathbf{x}_2$.

---

Using Theorem 4.3 with the parameter setting

$$T = \Theta\left(\frac{\sqrt{d}G_f G_g R}{\delta\epsilon^2} + \frac{\sqrt{d}G_f^2 G_g^2}{\epsilon^2}\right), \quad b_f = \Theta\left(\frac{dG_f^2 G_g^2}{\epsilon^2}\right), \quad b_g = b'_g = \Theta\left(\frac{d^2 G_f^2 \sigma_0^2}{\delta^2\epsilon^2}\right),$$

we obtain the following oracle complexity result for Algorithm 2.

**Corollary 4.4.** *Under Assumption 3.1, 3.3 and 3.4, the GFCOM⁺ algorithm (Algorithm 2) requires at most $\mathcal{O}\big(d^{3.5}G_f^4 G_g^2 \sigma_0^2 R\delta^{-3}\epsilon^{-5} + d^{3.5}G_f^5 G_g^3 \sigma_0^2 \delta^{-2}\epsilon^{-5}\big)$ stochastic zeroth-order function query calls to obtain a $(\delta, \epsilon)$-Goldstein stationary point of $\Phi$.*

For both Theorem 4.1 and 4.3, we take $c = 1$ according to Lemma 8 of Duchi et al. [36].

### 4.3 Discussion

In Algorithm 2, we only apply the variance reduction technique to the outer function $f(\cdot)$ to accelerate our algorithm. In contrast, existing methods [2, 7, 22] for nonconvex smooth SCO also apply the technique on the inner function estimator to obtain an improved complexity bound. Here we briefly discuss the cause that leads to such a difference. For smooth optimization, the main intuition of the variance reduction technique is to establish a connection between the bound of the mean-square error term $\mathbb{E}\big[\|\mathbf{v}_t - \nabla\Phi(\mathbf{x}_t))\|^2\big]$ and the expected distance of iterates at successive iterations $\mathbb{E}[\|\mathbf{x}_t - \mathbf{x}_{t-1}\|^2]$, which diminishes asymptotically. In our algorithm, we exploit the randomized smoothing with perturbed iterates $\mathbf{x}_t \pm \delta\mathbf{w}_{t,j}$ to approximate the gradient of the smoothed surrogate function $\Phi_\delta(\mathbf{x}_t)$. If we apply the variance reduction to the inner function estimator, the mean-square error $\mathbb{E}[\|\mathbf{v}_t - \nabla\Phi_\delta(\mathbf{x}_t))\|^2]$ is bounded by the expected distance of the perturbed iterates at successive iterations $\mathbb{E}[\|\mathbf{x}_t - \mathbf{x}_{t-1} \pm \delta(\mathbf{w}_{t,j} - \mathbf{w}_{t-1,j})\|^2]$, which does not vanish asymptotically.

## 5 Extensions to Convex Nonsmooth SCO

In this section, we extend the result in Section 4.1 to study the convex nonsmooth SCO problem. Firstly, we introduce an additional assumption as follows.

**Assumption 5.1.** *We suppose the function $f(\mathbf{x})$ is convex and non-decreasing, $G(\mathbf{x}; \boldsymbol{\zeta})$ is convex for any given $\boldsymbol{\zeta}$, and the solution set $\mathcal{X}^* = \arg\min_{\mathbf{x}\in\mathbb{R}^d} \Phi(\mathbf{x})$ is non-empty.*

From this assumption and Section 3.2.4 by Boyd and Vandenberghe [38], we can deduce that $\Phi(\mathbf{x})$ is a convex function. We will show that stochastic algorithms obtain an improved convergence rate for the nonsmooth SCO problem with Assumption 5.1. To obtain a $(\delta, \epsilon)$-Goldstein stationary point of the problem, we propose a two-phase gradient-free stochastic method called warm-started GFCOM (WS-GFCOM) in Algorithm 3. The intuition is that we use the GFCOM method to get a sufficiently small sub-optimality in the first phase, and then we apply the proposed methods in Section 4.1 to find the stationary point in the second phase. We remark that using the GFCOM method for the first phase is justified by the observation that it can achieve optimal convergence rate in terms of the sub-optimality of function value gap given the access to the exact function value $g(\mathbf{x})$ [39].

### 5.1 Convergence Analysis

In this subsection, we consider the complexity analysis of the proposed WS-GFCOM method. Let $\hat{R} \triangleq \min_{\mathbf{x}\in\mathcal{X}^*} \|x - x_0\|$. We characterize the convergence rate of the WS-GFCOM method for the convex nonsmooth SCO problem at the first phase with the following theorem.

**Theorem 5.2.** *Under Assumption 3.1, 3.3 and 5.1, running the WS-GFCOM algorithm (Algorithm 3) with parameters $\eta_0 = \Theta\big(\hat{R}/(G_f G_g \sqrt{dT_0})\big)$, $T_0 = \Theta\big(dG_f^2 G_g^2 \hat{R}^2 \rho^{-2}\big)$, $b_{g,0} = \Theta\big(G_f^2 \sigma_0^2 \rho^{-2}\big)$ and $\delta = \Theta\big(\rho G_f^{-1} G_g^{-1}\big)$, then the output $\mathbf{x}_1$ satisfies $\mathbb{E}[\Phi(\mathbf{x}_1) - \Phi^*] \leq \rho$. In addition, the total zeroth-order stochastic oracle complexity is at most $\mathcal{O}\big(dG_f^4 G_g^2 \sigma_0^2 \hat{R}^2 \rho^{-4}\big)$.*

Theorem 5.2 implies that the initial suboptimality at the beginning of the second phase is bounded by $\rho$. Consequently, the total complexity of WS-GFCOM[2] (Algorithm 3 with Option I) is bounded by $\mathcal{O}\big(dG_f^4 G_g^2 \sigma_0^2 \hat{R}^2 \rho^{-4} + d^{3.5} G_f^5 G_g^3 \sigma_0^2 \rho \delta^{-3} \epsilon^{-6} + d^{3.5} G_f^6 G_g^4 \sigma_0^2 \delta^{-2} \epsilon^{-6}\big)$. An appropriate choice of $\rho$ leads to the following oracle complexity of Algorithm 3 with Option I.

**Corollary 5.3.** *Under Assumption 3.1, 3.3 and 5.1, the WS-GFCOM[2] algorithm (Algorithm 3 with Option I) requires at most $\mathcal{O}\big(d^3 G_f^{4.8} G_g^{2.8} \sigma_0^2 \hat{R}^{0.4} \delta^{-2.4} \epsilon^{-4.8} + d^{3.5} G_f^6 G_g^4 \sigma_0^2 \delta^{-2} \epsilon^{-6}\big)$ stochastic zeroth-order function query calls to obtain a $(\delta, \epsilon)$-Goldstein stationary point of $\Phi$.*

With a similar deduction, we can show that using the GFCOM⁺ for the second phase can obtain an improved complexity bound. The following theorem shows the oracle complexity of Algorithm 3 with Option II.

**Corollary 5.4.** *Under Assumption 3.1, 3.3 and 5.1, the WS-GFCOM⁺ algorithm (Algorithm 3 with Option II) requires at most $\mathcal{O}\big(d^3 G_f^4 G_g^2 \sigma_0^2 \hat{R}^{0.4} \delta^{-2.4} \epsilon^{-4} + d^{3.5} G_f^5 G_g^3 \sigma_0^2 \delta^{-2} \epsilon^{-5}\big)$ stochastic zeroth-order function query calls to obtain a $(\delta, \epsilon)$-Goldstein stationary point of $\Phi$.*

# 6  Experiments

We compare the proposed methods GFCOM and GFCOM⁺ with a Kiefer-Wolfowitz style zeroth-order baseline method [2, 40]. In particular, the baseline gradient estimator is defined as

$$\mathbf{v}_t = \frac{1}{b_f} \sum_{j \in [b_f]} \frac{d}{2\delta} \left( F(\mathbf{y}_{t,j}; \boldsymbol{\xi}_{t,j}) - F(\mathbf{z}_{t,j}; \boldsymbol{\xi}'_{t,j}) \right),$$

where $\mathbf{y}_{t,j}$ and $\mathbf{z}_{t,j}$ are function estimators defined in Eq. (3). $\boldsymbol{\xi}_{t,j}$ and $\boldsymbol{\xi}'_{t,j}$ are independent random variables. We test all the methods on the nonconvex penalized risk-averse portfolio management problem and a reinforcement learning (RL) problem. We set $\delta = 0.1$ for the GFCOM and GFCOM⁺ methods.

## 6.1  Nonconvex Penalized Portfolio Management

We consider the portfolio management problem with capped-$\ell_1$ regularizer [41]. Let $\mathbf{x}$ denote the investment quantity corresponding to $N$ assets and $\mathbf{r}_t \in \mathbb{R}^N$ denote the returns of $N$ assets at timestamp $t$. We can formulate the portfolio management problem as the following nonsmooth compositional optimization problem

$$\min_{\mathbf{x} \in \mathbb{R}^N} -\frac{1}{T} \sum_{t=1}^{T} \langle \mathbf{r}_t, \mathbf{x} \rangle + \frac{1}{T} \sum_{i=1}^{T} \left( \langle \mathbf{r}_t, \mathbf{x} \rangle - \frac{1}{T} \sum_{s=1}^{T} \langle \mathbf{r}_s, \mathbf{x} \rangle \right)^2 + \beta(\mathbf{x}), \tag{6}$$

where $\beta(\mathbf{x}) = \lambda \sum_{i=1}^{N} \min\{|x_i|, \alpha\}$ and $\lambda, \alpha > 0$ are tunable hyperparameters. Specifically, the inner function $G(\mathbf{x}; \xi)$ and outer function $F(\mathbf{w}; \zeta)$ can be formulated as

$$G(\mathbf{x}; \boldsymbol{\xi}) = [x_1, \ldots, x_N, \langle r_\xi, \mathbf{x} \rangle]^\top$$

and

$$F(\mathbf{w}; \zeta) = -\langle \mathbf{r}_\zeta, w_{[N]} \rangle + (\langle \mathbf{r}_\zeta, \mathbf{w}_{[N]} \rangle - \mathbf{w}_{N+1})^2 + \beta(\mathbf{x}).$$

Both random variables $\boldsymbol{\xi}$ and $\boldsymbol{\zeta}$ are uniformly sampled from $\{1, \ldots, T\}$. We choose $\lambda = 10^{-5}$ and $\alpha = 2$ in our experiments. The goal of the Problem (6) is to maximize the return while controlling the variance of the portfolio.

We compare all the methods on 6 different portfolio datasets formed on Size and Operating Profitability[2]. For all algorithms, we tune the stepsize among $\{1 \times 10^{-5}, 3 \times 10^{-5}, \ldots, 1 \times 10^{-3}, 3 \times 10^{-3}\}$.

---

[2]http://mba.tuck.dartmouth.edu/pages/faculty/ken.french/data_library.html

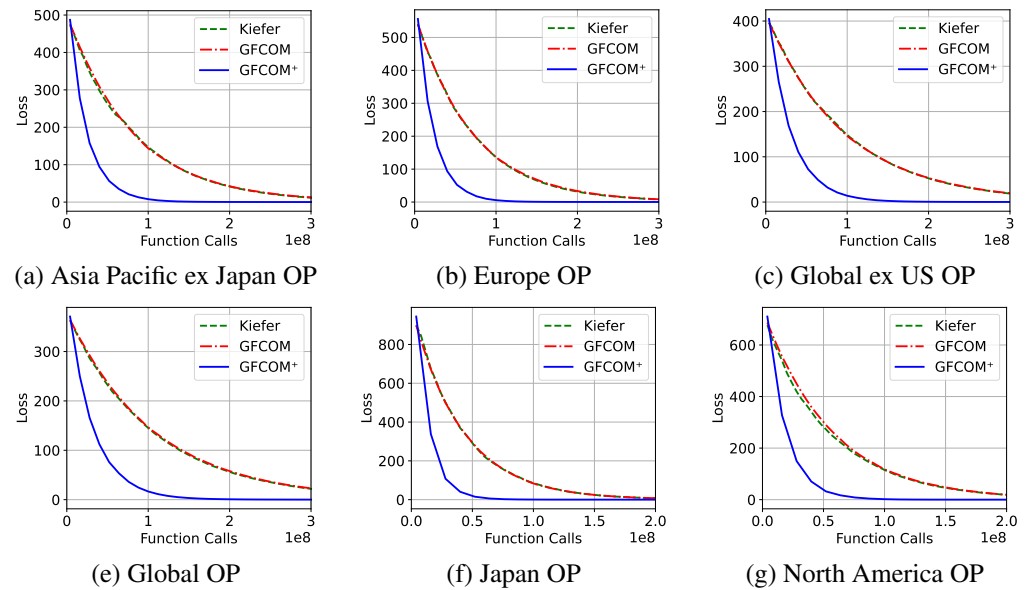

Figure 1: We present the loss vs. complexity on several portfolio management datasets. The plot of GFCOM and the Kiefer-Wolfowitz method are overlapped as their performance are close to each other.

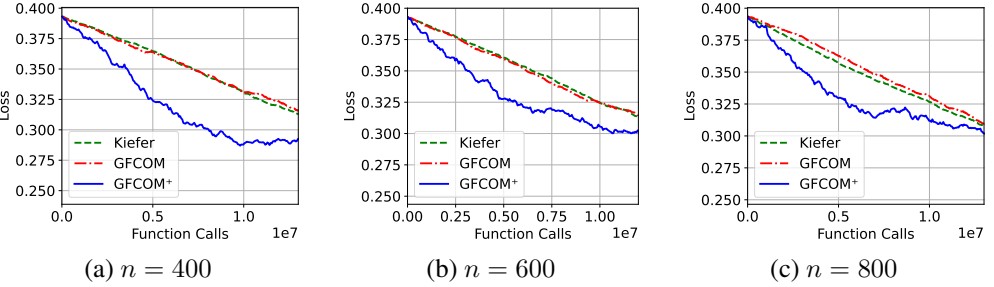

Figure 2: For the RL task, we present the loss vs. complexity on datasets with states of different sizes.

We choose the mini-batch size $b_f = b_g = 1000$. In addition, we set $b'_f = 100$, $b'_g = 1000$ and $m = b_f/b'_f = 10$ for the GFCOM$^+$ algorithm. Figure 1 shows that the GFCOM$^+$ algorithm converges much faster than the GFCOM and the baseline method across all datasets.

## 6.2 Application to Reinforcement Learning

We demonstrate an experiment on RL and verify the effectiveness of the proposed methods on value function evaluation. Let $V^\pi(s)$ be the value function of a state $s$ under a policy $\pi$ for all state $s \in \mathcal{S}$ where $|\mathcal{S}| = n$. Let $r_{s',s}$ be the reward transition from $s'$ to $s$, and $\gamma > 0$ is a discounting factor. Furthermore, we assume that the value of each state can be parameterized as a linear map of some feature map $\psi_s \in \mathbb{R}^d$ of the state $s$ such that $V^\pi(s) = \langle \psi_s, \mathbf{w} \rangle$. Then we formulate the RL problem as a Bellman residual minimization problem

$$\min_{\mathbf{w} \in \mathbb{R}^d} \sum_{s=1}^{n} h\left( \langle \psi_s, \mathbf{w} \rangle - \sum_{s'} P_{ss'}(r_{s,s'} + \gamma \langle \psi_{s'}, \mathbf{w} \rangle) \right),$$

where $P_{ss'}$ is the probability transition matrix and $h(x) = 1 - \exp(-|x|/\sigma)$ is a nonconvex nonsmooth loss which is more robust to adversarial outliers than the squared loss [42, 43]. Specifically, the inner function $G(\mathbf{x}; \boldsymbol{\xi})$ and outer function $F(\mathbf{w}; \boldsymbol{\zeta})$ can be formulated as

$$G(\mathbf{w}; \boldsymbol{\xi}) = [\langle \psi_1, \mathbf{w} \rangle, r_{1,\boldsymbol{\xi}_1} + \gamma \langle \psi_{\boldsymbol{\xi}_1}, \mathbf{w} \rangle, \ldots, \langle \psi_n, \mathbf{w} \rangle, r_{n,\boldsymbol{\xi}_n} + \gamma \langle \psi_{\boldsymbol{\xi}_n}, \mathbf{w} \rangle]^\top$$

and

$$F(\mathbf{z}; \boldsymbol{\zeta}) = h(z_{2\zeta} - z_{2\zeta+1}).$$

In the above formulation, each $\boldsymbol{\xi}_i$ is uniformly sampled from $\{P_{i1}, \ldots, P_{in}\}$ and $\zeta$ is uniformly sampled from $\{1, \ldots, T\}$. We follow a similar experiment setup by Yuan et al. [22]. Specifically, we generate a Markov decision process with different numbers of states $n \in \{400, 600, 800\}$ and 10 actions at each state. The transition probability matrix is generated from the uniform distribution from $[0, 1]$. In addition, the rewards are sampled uniformly from $[0, 1]$. In terms of hyperparameter setting, we choose $b_f = b_g = 100$ for all algorithms. In addition, we set $b'_f = 10$, $b'_g = 100$ and $m = b_f/b'_f = 10$ for the GFCOM$^+$ algorithm. For other hyperparameters, we use the same setting for the portfolio management problem. The experimental results in Figure 2 show that the GFCOM$^+$ significantly outperforms other methods.

## 7   Conclusion

In this work, we propose novel zeroth-order algorithms for nonconvex nonsmooth stochastic compositional optimization. We present the non-asymptotic convergence rate of the proposed algorithms for obtaining a $(\delta, \epsilon)$-Goldstein point of the problem. Furthermore, we extend our methods with a warm-start phase to solve the convex nonsmooth SCO problem with improved convergence guarantees. We conduct numerical experiments on portfolio management and reinforcement learning problems to demonstrate the effectiveness of the proposed algorithms.

In future work, it is interesting to study the lower bound of the zeroth-order algorithms on nonconvex nonsmooth SCO. It is also interesting to investigate whether the complexity bound of zeroth-order algorithms can be further improved.

## Acknowledgement

This research/project is supported by the National Research Foundation, Singapore under its AI Singapore Programme (AISG Award No: AISG2-PhD-2023-08-043T-J). This research is part of the programme DesCartes and is supported by the National Research Foundation, Prime Minister's Office, Singapore under its Campus for Research Excellence and Technological Enterprise (CREATE) programme. Luo Luo is supported by National Natural Science Foundation of China (No. 62206058), Shanghai Sailing Program (22YF1402900), Shanghai Basic Research Program (23JC1401000), and the Major Key Project of PCL under Grant PCL2024A06.

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

The appendix is organized as follows. Section A introduces supporting lemmas that are essential for the analysis of the proposed gradient-free SCO methods. Section B proves the convergence rate of the GFCOM method introduced in Section 4. Section C proves the convergence rate of the GFCOM+ method which enjoys a better function oracle call complexity. Section D provides the convergence analysis of the WS-GFCOM$^2$ and WS-GFCOM+ proposed in Section 5.

## A  Supporting Lemmas

Throughout the work, we define the variable $g_t(\mathbf{x}) = \frac{1}{b_g} \sum_{i \in [b_g]} G(\mathbf{x}; \boldsymbol{\zeta}_{t,i})$ for the GFCOM algorithm, and we denote

$$g_t(\mathbf{x}) = \begin{cases} \frac{1}{b_g} \sum_{i \in [b_g]} G(\mathbf{x}; \boldsymbol{\zeta}_{t,i}), & t \bmod m = 0 \\ \frac{1}{b'_g} \sum_{i \in [b'_g]} G(\mathbf{x}; \boldsymbol{\zeta}_{t,i}), & \text{Otherwise} \end{cases} \tag{7}$$

for the GFCOM+ method. Now we introduce an important lemma which is useful for the analysis of both GFCOM and GFCOM+ algorithms.

**Lemma A.1.** *Under Assumption 3.1 and 3.3, for both Algorithms 1 and 2 it holds that*

$$\mathbb{E}[(f \circ g)_\delta(\mathbf{x}_{t+1}) - (f \circ g)_\delta(\mathbf{x}_t)]$$

$$\leq -\frac{\eta}{2} \mathbb{E}\left[\|\nabla(f \circ g)_\delta(\mathbf{x}_t)\|^2\right] - \left(\frac{\eta}{2} - \frac{c\eta^2 G_f G_g \sqrt{d}}{2\delta}\right) \mathbb{E}\left[\|\mathbf{v}_t\|^2\right] + \frac{\eta}{2} \mathbb{E}\left[\|\mathbf{v}_t - \nabla(f \circ g)_\delta(\mathbf{x}_t)\|^2\right].$$

*Proof.* From the smoothness of $\Phi_\delta = (f \circ g)_\delta$, we have

$$(f \circ g)_\delta(\mathbf{x}_{t+1}) - (f \circ g)_\delta(\mathbf{x}_t)$$

$$\leq \langle \nabla(f \circ g)_\delta(\mathbf{x}_t), \mathbf{x}_{t+1} - \mathbf{x}_t \rangle + \frac{cG_f G_g \sqrt{d}}{2\delta} \|\mathbf{x}_{t+1} - \mathbf{x}_t\|^2$$

$$= -\eta[\langle \nabla(f \circ g)_\delta(\mathbf{x}_t), \mathbf{v}_t \rangle + \frac{c\eta^2 G_f G_g \sqrt{d}}{2\delta} \|\mathbf{v}_t\|^2$$

$$= -\frac{\eta}{2} \|\nabla(f \circ g)_\delta(\mathbf{x}_t)\|^2 - \left(\frac{\eta}{2} - \frac{c\eta^2 G_f G_g \sqrt{d}}{2\delta}\right) \|\mathbf{v}_t\|^2 + \frac{\eta}{2} \|\mathbf{v}_t - \nabla(f \circ g)_\delta(\mathbf{x}_t)\|^2.$$

Taking expectations on both sides of the inequality, we get the desired result. □

## B  Convergence Analysis of Algorithm 1

Before giving the analysis of the convergence rate of Algorithm 1, we first present the bound of the mean-square error term $\mathbb{E}\left[\|\mathbf{v}_t - \nabla\Phi_\delta(\mathbf{x}_t)\|^2\right]$.

**Lemma B.1.** *Under Assumption 3.1 and 3.3, for Algorithm 1 it holds that*

$$\mathbb{E}\left[\|\mathbf{v}_t - \nabla\Phi_\delta(\mathbf{x}_t)\|^2\right] \leq \frac{2d^2 G_f^2 \sigma_0^2}{\delta^2 b_g} + \frac{32\sqrt{2\pi} d G_f^2 G_g^2}{b_f}$$

*Proof.* Recall that $\mathbf{v}_t = \frac{1}{b_f} \sum_{j \in [b_f]} \frac{d}{2\delta}(F(\mathbf{y}_{t,j}; \boldsymbol{\xi}_{t,j}) - F(\mathbf{z}_{t,j}; \boldsymbol{\xi}_{t,j}))$, we have

$$\mathbb{E}\left[\|\mathbf{v}_t - \nabla\Phi_\delta(x_t)\|^2\right]$$

$$\leq 2\mathbb{E}\left[\left\|\mathbf{v}_t - \frac{1}{b_f} \sum_{j \in [b_f]} \frac{d}{2\delta}(F(g(\mathbf{x}_t + \delta\mathbf{w}_{t,j}); \boldsymbol{\xi}_{t,j}) - F(g(\mathbf{x}_t - \delta\mathbf{w}_{t,j}); \boldsymbol{\xi}_{t,j}))\right\|^2\right]$$

$$+ 2\mathbb{E}\left[\left\|\frac{1}{b_f} \sum_{j \in [b_f]} \frac{d}{2\delta}(F(g(\mathbf{x}_t + \delta\mathbf{w}_{t,j}); \boldsymbol{\xi}_{t,j}) - F(g(\mathbf{x}_t - \delta\mathbf{w}_{t,j}); \boldsymbol{\xi}_{t,j})) - \nabla(f \circ g)_\delta(x_t)\right\|^2\right]$$

$$
\leq 2 \left( \frac{d^2}{2\delta^2 b_f} \sum_{j \in [b_f]} \mathbb{E} \left[ \| F(g(\mathbf{x}_t + \delta \mathbf{w}_{t,j}); \boldsymbol{\xi}_{t,j}) - F(g_t(\mathbf{x}_t + \delta \mathbf{w}_{t,j}); \boldsymbol{\xi}_{t,j}) \|^2 \right] \right.
$$

$$
\left. + \frac{d^2}{2\delta^2 b_f} \sum_{j \in [b_f]} \mathbb{E} \left[ \| F(g(\mathbf{x}_t - \delta \mathbf{w}_{t,j}); \boldsymbol{\xi}_{t,j}) - F(g_t(\mathbf{x}_t - \delta \mathbf{w}_{t,j}); \boldsymbol{\xi}_{t,j}) \|^2 \right] \right)
$$

$$
+ \frac{32\sqrt{2\pi} d G_f^2 G_g^2}{b_f}
$$

$$
\leq \frac{2d^2 G_f^2 \sigma_0^2}{\delta^2 b_g} + \frac{32\sqrt{2\pi} d G_f^2 G_g^2}{b_f}.
$$

The first inequality is due to $\| \mathbf{a} + \mathbf{b} \|^2 \leq 2 \| \mathbf{a} \|^2 + 2 \| \mathbf{b} \|^2$ for any $\mathbf{a}, \mathbf{b} \in \mathbb{R}^d$. The second inequality is due to Lemma 3.8. The last inequality follows from the $G_f$-Lipchitzness of $f(\cdot)$ and Assumption 3.3. $\qquad\square$

We present the formal proof of Theorem 4.1 and Corollary 4.2 below.

### B.1 Proof of Theorem 4.1

*Proof.* If we take $\eta = \frac{\delta}{c G_f G_g \sqrt{d}}$ and rearrange the formula in Lemma A.1, we have

$$
\frac{\eta}{2} \mathbb{E}[\| \nabla (f \circ g)_\delta (\mathbf{x}_t) \|^2] \leq \mathbb{E}[(f \circ g)_\delta (\mathbf{x}_t) - (f \circ g)_\delta (\mathbf{x}_{t+1})] + \frac{\eta}{2} \mathbb{E}[\| \mathbf{v}_t - \nabla (f \circ g)_\delta (\mathbf{x}_t) \|^2].
$$

Sum the above inequality from $t = 0$ to $T - 1$ and divide both sides by $\frac{\eta T}{2}$, we have

$$
\frac{1}{T} \sum_{t=0}^{T-1} \mathbb{E} \left[ \| \nabla (f \circ g)_\delta (\mathbf{x}_t) \|^2 \right] \leq \frac{2c G_f G_g \sqrt{d} \mathbb{E} \left[ (f \circ g)_\delta (\mathbf{x}_0) - (f \circ g)_\delta (\mathbf{x}_T) \right]}{\delta T}
$$

$$
+ \frac{32\sqrt{2\pi} d G_f^2 G_g^2}{b_f} + \frac{2d^2 G_f^2 \sigma_0^2}{\delta^2 b_g}.
$$

In addition, by Lemma 3.7, we have

$$
\mathbb{E} \left[ (f \circ g)_\delta (\mathbf{x}_0) - (f \circ g)_\delta (\mathbf{x}_T) \right] \leq \mathbb{E}[(f \circ g)(\mathbf{x}_0) - (f \circ g)(\mathbf{x}_T)] + 2 G_f G_g \delta.
$$

Combining the last two inequalities, we get the desired result. $\qquad\square$

### B.2 Proof of Corollary 4.2

*Proof.* The total zeroth-order oracle calls can be bounded by

$$
\mathcal{O}(T b_f b_g)
$$

$$
= \mathcal{O} \left( \left( \frac{G_f G_g \sqrt{d} R}{\delta \epsilon^2} + \frac{G_f^2 G_g^2 \sqrt{d}}{\epsilon^2} \right) \cdot \frac{d G_f^2 G_g^2}{\epsilon^2} \cdot \frac{d^2 G_f^2 \sigma_0^2}{\delta^2 \epsilon^2} \right)
$$

$$
= \mathcal{O} \left( \frac{d^{3.5} G_f^5 G_g^3 \sigma_0^2 R}{\delta^3 \epsilon^6} + \frac{d^{3.5} G_f^6 G_g^4 \sigma_0^2}{\delta^2 \epsilon^6} \right).
$$

$\qquad\square$

## C Convergence Analysis of Algorithm 2

In this section, we consider the formal proof of the convergence rate of the GFCOM⁺ method. First, we introduce the following lemma to bound the mean-square error between the recursive gradient estimator and the gradient of the surrogate function.

**Lemma C.1.** *Let $n_t = \lfloor t/m \rfloor \, m$. Under Assumption 3.1 and 3.3, for Algorithm 2 it holds that*

$$\mathbb{E}\left[\|\mathbf{v}_t - \nabla(f \circ g)_\delta(\mathbf{x}_t)\|^2\right]$$

$$\leq \frac{10d^2 G_f^2 G_g^2 \eta^2}{\delta^2 b_f'} \sum_{i=n_t}^{t} \mathbb{E}\left[\|\mathbf{v}_i\|^2\right] + \frac{10md^2 G_f^2 \sigma_0^2}{\delta^2 b_f' b_g'} + \frac{32\sqrt{2\pi} d G_f^2 G_g^2}{b_f}$$

$$+ \frac{2cd^2 G_f^2 \sigma_0^2}{\delta^2 b_g} + \frac{2cd^2 G_f^2 \sigma_0^2}{\delta^2 b_g'}.$$

*Proof.* By $\|\mathbf{a} + \mathbf{b}\|^2 \leq 2\|\mathbf{a}\|^2 + 2\|\mathbf{b}\|^2$, we can infer that

$$\mathbb{E}\left[\|\mathbf{v}_t - \nabla(f \circ g)_\delta(\mathbf{x}_t)\|^2\right]$$

$$\leq 2\mathbb{E}\left[\|\mathbf{v}_t - \nabla(f \circ g_t)_\delta(\mathbf{x}_t)\|^2\right] + 2\mathbb{E}\left[\|\nabla(f \circ g_t)_\delta(\mathbf{x}_t) - \nabla(f \circ g)_\delta(\mathbf{x}_t)\|^2\right].$$

To bound the first term of R.H.S., we have

$$\mathbb{E}[\|\mathbf{v}_t - \nabla(f \circ g_t)_\delta(\mathbf{x}_t)\|^2]$$

$$= \mathbb{E}\left[\left\|\frac{1}{b_f'}\sum_{j \in [b_f']}\left[\frac{d}{2\delta}(F(\mathbf{y}_{t,j};\boldsymbol{\xi}_{t,j}) - F(\mathbf{z}_{t,j};\boldsymbol{\xi}_{t,j}))\mathbf{w}_{t,j} - \frac{d}{2\delta}(F(\mathbf{y}_{t-1,j};\boldsymbol{\xi}_{t,j}) - F(\mathbf{z}_{t-1,j};\boldsymbol{\xi}_{t,j}))\mathbf{w}_{t,j}\right]\right.\right.$$

$$\left.\left. + \mathbf{v}_{t-1} - \nabla(f \circ g_t)_\delta(\mathbf{x}_t)\right\|^2\right]$$

$$= \mathbb{E}\left[\left\|\frac{1}{b_f'}\sum_{j \in [b_f']}\left[\frac{d}{2\delta}(F(\mathbf{y}_{t,j};\boldsymbol{\xi}_{t,j}) - F(\mathbf{z}_{t,j};\boldsymbol{\xi}_{t,j}))\mathbf{w}_{t,j} - \frac{d}{2\delta}(F(\mathbf{y}_{t-1,j};\boldsymbol{\xi}_{t,j}) - F(\mathbf{z}_{t-1,j};\boldsymbol{\xi}_{t,j}))\mathbf{w}_{t,j}\right]\right.\right.$$

$$\left.\left. - (\nabla(f \circ g_t)_\delta(\mathbf{x}_t) - \nabla(f \circ g_{t-1})_\delta(\mathbf{x}_{t-1}))\right\|^2\right] + \mathbb{E}\left[\|\mathbf{v}_{t-1} - \nabla(f \circ g_{t-1})_\delta(\mathbf{x}_{t-1})\|^2\right]$$

$$\leq \frac{1}{b_f'^2}\sum_{j \in [b_f']}\mathbb{E}\left[\left\|\frac{d}{2\delta}(F(\mathbf{y}_{t,j};\boldsymbol{\xi}_{t,j}) - F(\mathbf{z}_{t,j};\boldsymbol{\xi}_{t,j}))\mathbf{w}_{t,j} - \frac{d}{2\delta}(F(\mathbf{y}_{t-1,j};\boldsymbol{\xi}_{t,j}) - F(\mathbf{z}_{t-1,j};\boldsymbol{\xi}_{t,j}))\mathbf{w}_{t,j}\right\|^2\right]$$

$$+ \mathbb{E}\left[\|\mathbf{v}_{t-1} - \nabla(f \circ g_{t-1})_\delta(\mathbf{x}_{t-1})\|^2\right].$$

The second equality follows from Lemma 3.8. The last inequality is due to $\mathbb{E}[\|\mathbf{x} - \mathbb{E}[\mathbf{x}]\|^2] \leq \mathbb{E}[\|\mathbf{x}\|^2]$. Observe that

$$\mathbb{E}\left[\left\|\frac{d}{2\delta}(F(\mathbf{y}_{t,j};\boldsymbol{\xi}_{t,j}) - F(\mathbf{z}_{t,j};\boldsymbol{\xi}_{t,j}))\mathbf{w}_{t,j} - \frac{d}{2\delta}(F(\mathbf{y}_{t-1,j};\boldsymbol{\xi}_{t,j}) - F(\mathbf{z}_{t-1,j};\boldsymbol{\xi}_{t,j}))\mathbf{w}_{t,j}\right\|^2\right]$$

$$\leq 5\mathbb{E}\left[\left\|\frac{d}{2\delta}(F(g(\mathbf{x}_t + \delta\mathbf{w}_{t,j});\boldsymbol{\xi}_{t,j}) - F(g(\mathbf{x}_t - \delta\mathbf{w}_{t,j});\boldsymbol{\xi}_{t,j}) - \right.\right.$$

$$\left.\left. (F(g(\mathbf{x}_{t-1} + \delta\mathbf{w}_{t,j});\boldsymbol{\xi}_{t,j}) - F(g(\mathbf{x}_{t-1} - \delta\mathbf{w}_{t,j});\boldsymbol{\xi}_{t,j})))\mathbf{w}_{t,j}\right\|^2\right]$$

$$+ 5\mathbb{E}\left[\left\|\frac{d}{2\delta}(F(g(\mathbf{x}_t + \delta\mathbf{w}_{t,j});\boldsymbol{\xi}_{t,j}) - F(\mathbf{y}_{t,j};\boldsymbol{\xi}_{t,j}))\mathbf{w}_{t,j}\right\|^2\right]$$

$$+ 5\mathbb{E}\left[\left\|\frac{d}{2\delta}(F(g(\mathbf{x}_t - \delta\mathbf{w}_{t,j});\boldsymbol{\xi}_{t,j}) - F(\mathbf{z}_{t,j};\boldsymbol{\xi}_{t,j}))\mathbf{w}_{t,j}\right\|^2\right]$$

$$+ 5\mathbb{E}\left[\left\|\frac{d}{2\delta}(F(g(\mathbf{x}_{t-1} + \delta\mathbf{w}_{t,j});\boldsymbol{\xi}_{t,j}) - F(\mathbf{y}_{t-1,j};\boldsymbol{\xi}_{t,j}))\mathbf{w}_{t,j}\right\|^2\right]$$

$$+ 5\mathbb{E}\left[\left\|\frac{d}{2\delta}(F(g(\mathbf{x}_{t-1} - \delta\mathbf{w}_{t,j});\boldsymbol{\xi}_{t,j}) - F(\mathbf{z}_{t-1,j};\boldsymbol{\xi}_{t,j}))\mathbf{w}_{t,j}\right\|^2\right]$$

$$\leq \frac{5d^2 G_f^2 G_g^2}{\delta^2} \mathbb{E}\left[\|\mathbf{x}_t - \mathbf{x}_{t-1}\|^2\right] + \frac{5d^2 G_f^2 \sigma_0^2}{\delta^2 b_g'}$$

$$= \frac{5d^2 G_f^2 G_g^2 \eta^2}{\delta^2} \mathbb{E}\left[\|\mathbf{v}_{t-1}\|^2\right] + \frac{5d^2 G_f^2 \sigma_0^2}{\delta^2 b_g'}.$$

The first inequality is due to $\|a_1 + \cdots + a_n\|^2 \leq n\|a_1\|^2 + \cdots + n\|a_n\|^2$. The second inequality is due to the Lipschitzness of both inner and outer functions with Assumption 3.3. Consequently, one has

$$\mathbb{E}\left[\|\mathbf{v}_t - \nabla(f \circ g_t)_\delta(\mathbf{x}_t)\|^2\right]$$

$$\leq \frac{5d^2 G_f^2 G_g^2 \eta^2}{\delta^2 b_f'} \mathbb{E}\left[\|\mathbf{v}_{t-1}\|^2\right] + \frac{5d^2 G_f^2 \sigma_0^2}{\delta^2 b_f' b_g'} + \mathbb{E}\left[\|\mathbf{v}_{t-1} - \nabla(f \circ g_{t-1})_\delta(\mathbf{x}_{t-1})\|^2\right]$$

$$\leq \frac{5d^2 G_f^2 G_g^2 \eta^2}{\delta^2 b_f'} \sum_{i=n_t}^{t} \mathbb{E}\left[\|\mathbf{v}_i\|^2\right] + \frac{5m d^2 G_f^2 \sigma_0^2}{\delta^2 b_f' b_g'} + \frac{16\sqrt{2\pi} d G_f^2 G_g^2}{b_f}.$$

The last inequality follows from Lemma 3.8. In addition, for $t \bmod m = 0$ we can bound

$$\mathbb{E}\left[\|\nabla(f \circ g_t)_\delta(\mathbf{x}_t) - \nabla(f \circ g)_\delta(\mathbf{x}_t)\|^2\right]$$

$$= \mathbb{E}\left[\left\|\mathbb{E}_{\mathbf{u}}\left[\frac{d}{\delta}\left((f \circ g_t)(\mathbf{x}_t + \delta\mathbf{u}) - (f \circ g)(\mathbf{x}_t + \delta\mathbf{u})\right)\mathbf{u}\right]\right\|^2\right]$$

$$\leq \frac{d^2}{\delta^2} \mathbb{E}_{\mathbf{u}}\left[\|(f \circ g_t)(\mathbf{x}_t + \delta\mathbf{u}) - (f \circ g)(\mathbf{x}_t + \delta\mathbf{u})\|^2 \|\mathbf{u}\|^2\right]$$

$$\leq \frac{cd^2 G_f^2 \sigma_0^2}{\delta^2 b_g}.$$

The last inequality follows from the Lipschitzness of $f$ and Assumption 3.3. Similarly, for $t \bmod m \neq 0$ we can bound

$$\mathbb{E}\left[\|\nabla(f \circ g_t)_\delta(\mathbf{x}_t) - \nabla(f \circ g)_\delta(\mathbf{x}_t)\|^2\right] \leq \frac{cd^2 G_f^2 \sigma_0^2}{\delta^2 b_g'}.$$

Putting everything together, we get the desired bound. $\qquad\square$

We present the formal proof of Theorem 4.3 and Corollary 4.4 below.

## C.1 Proof of Theorem 4.3

*Proof.* Rearrange the terms in Lemma A.1, sum $t$ from 0 to $T-1$, and divide both sides by $\frac{\eta T}{2}$,

$$\frac{1}{T}\sum_{t=0}^{T-1} \mathbb{E}\left[\|\nabla(f \circ g)_\delta(\mathbf{x}_t)\|^2\right]$$

$$\leq \frac{2\mathbb{E}\left[(f \circ g)_\delta(\mathbf{x}_0) - 2(f \circ g)_\delta(\mathbf{x}_T)\right]}{\eta T} - \frac{1}{T}\left(1 - \frac{c\eta G_f G_g \sqrt{d}}{\delta}\right)\sum_{i=0}^{T-1}\mathbb{E}\left[\|\mathbf{v}_i\|^2\right]$$

$$+ \frac{1}{T}\sum_{t=0}^{T-1}\mathbb{E}\left[\|\mathbf{v}_t - \nabla(f \circ g)_\delta(\mathbf{x}_t)\|^2\right]$$

$$\leq \frac{2\mathbb{E}\left[(f \circ g)_\delta(\mathbf{x}_0) - 2(f \circ g)_\delta(\mathbf{x}_T)\right]}{\eta T} - \frac{1}{T}\left(1 - \frac{c\eta G_f G_g \sqrt{d}}{\delta} - \frac{10m d^2 G_f^2 G_g^2 \eta^2}{\delta^2 b_f'}\right)\sum_{i=0}^{T-1}\mathbb{E}\left[\|\mathbf{v}_i\|^2\right]$$

$$+ \frac{10m d^2 G_f^2 \sigma_0^2}{\delta^2 b_f' b_g'} + \frac{32\sqrt{2\pi} d G_f^2 G_g^2}{b_f} + \frac{2cd^2 G_f^2 \sigma_0^2}{\delta^2 b_g} + \frac{2cd^2 G_f^2 \sigma_0^2}{\delta^2 b_g'}.$$

The last inequality follows from Lemma C.1. If we choose hyperparameters as follows

$$\eta = \frac{\delta}{2cG_fG_g\sqrt{d}}, \quad b'_f = \Theta\left(\frac{dG_fG_g}{\epsilon}\right), \quad m = \Theta\left(\frac{G_fG_g}{\epsilon}\right),$$

Then we can deduce that $1 - \frac{c\eta G_fG_g\sqrt{d}}{\delta} - \frac{10md^2G_f^2G_g^2\eta^2}{\delta^2b'_f} \le 0$. By Lemma 3.7, we have

$$\mathbb{E}\left[(f \circ g)_\delta(\mathbf{x}_0) - (f \circ g)_\delta(\mathbf{x}_T)\right] \le \mathbb{E}[(f \circ g)(\mathbf{x}_0) - (f \circ g)(\mathbf{x}_T)] + 2G_fG_g\delta.$$

Therefore, we obtain the following result

$$\mathbb{E}\left[\|\nabla\Phi_\delta(\mathbf{x}_R)\|^2\right] = \mathcal{O}\left(\frac{\sqrt{d}G_fG_gR}{\delta T} + \frac{\sqrt{d}G_f^2G_g^2}{T} + \frac{dG_f^2G_g^2}{b_f} + \frac{d^2G_f^2\sigma_0^2}{\delta^2 b_g} + \frac{d^2G_f^2\sigma_0^2}{\delta^2 b'_g}\right).$$

$\square$

## C.2    Proof of Corollary 4.4

*Proof.* The total zeroth-order oracle calls can be bounded by

$$\mathcal{O}(Tb'_fb'_g + Tb_fb_g/m)$$
$$=\mathcal{O}\left(\left(\frac{\sqrt{d}G_fG_gR}{\delta\epsilon^2} + \frac{\sqrt{d}G_f^2G_g^2}{\epsilon^2}\right) \cdot \frac{dG_fG_g}{\epsilon} \cdot \frac{d^2G_f^2\sigma_0^2}{\delta^2\epsilon^2}\right)$$
$$=\mathcal{O}\left(\frac{d^{3.5}G_f^4G_g^2\sigma_0^2R}{\delta^3\epsilon^5} + \frac{d^{3.5}G_f^5G_g^3\sigma_0^2}{\delta^2\epsilon^5}\right).$$

$\square$

# D    Extensions to Convex Nonsmooth Functions

In this section, we present the formal proof of theorems presented in Section 5.

## D.1    Proof of Theorem 5.2

*Proof.* Since $G(\cdot; \boldsymbol{\zeta})$ is convex function, $g_t(\cdot)$ is also convex. Using the result of Section 3.2.4 of [38] and Assumption 5.1, we can deduce that $f \circ g_t$ is a convex function. Let $\mathbf{x}^* = \arg\min_{\mathbf{x}}(f \circ g)_\delta(\mathbf{x})$, then we have

$$\mathbb{E}[(f \circ g_t)_\delta(\mathbf{x}_t) - (f \circ g_t)_\delta(\mathbf{x}^*)]$$
$$\le\mathbb{E}[\langle\nabla(f \circ g_t)_\delta(\mathbf{x}_t), \mathbf{x}_t - \mathbf{x}^*\rangle]$$
$$=\mathbb{E}\left[\left\langle\frac{d}{2\delta}(F(g_t(\mathbf{x}_t + \delta\mathbf{u}), \boldsymbol{\xi}_t) - F(g_t(\mathbf{x}_t - \delta\mathbf{u}), \boldsymbol{\xi}_t)), \mathbf{x}_t - \mathbf{x}^*\right\rangle\right]$$
$$\le\frac{1}{\eta_0}\mathbb{E}\left[\langle\mathbf{x}_t - \mathbf{x}_{t+1}, \mathbf{x}_t - \mathbf{x}^*\rangle\right]$$
$$=\frac{1}{2\eta_0}\mathbb{E}\left[\|\mathbf{x}_t - \mathbf{x}^*\|^2 - \|\mathbf{x}_{t+1} - \mathbf{x}^*\|^2 + \|\mathbf{x}_t - \mathbf{x}_{t+1}\|^2\right]$$
$$\le\frac{1}{2\eta_0}\mathbb{E}\left[\|\mathbf{x}_t - \mathbf{x}^*\|^2 - \|\mathbf{x}_{t+1} - \mathbf{x}^*\|^2\right] + 8\sqrt{2\pi}dG_f^2G_g^2\eta_0.$$

The first inequality follows from the convexity of $f \circ g_t$. The last inequality is due to Lemma 3.8. Observe that for $\forall\mathbf{x} \in \mathbb{R}^d$,

$$|(f \circ g)_\delta(\mathbf{x}) - (f \circ g_t)_\delta(\mathbf{x})|$$
$$=|\mathbb{E}_\mathbf{u}[(f \circ g)(\mathbf{x} + \delta\mathbf{u})] - \mathbb{E}_\mathbf{u}[(f \circ g_t)(\mathbf{x} + \delta\mathbf{u})]|$$
$$=|\mathbb{E}_\mathbf{u}[(f \circ g)(\mathbf{x} + \delta\mathbf{u}) - (f \circ g_t)(\mathbf{x} + \delta\mathbf{u})]|$$
$$\le\mathbb{E}_\mathbf{u}[|(f \circ g)(\mathbf{x} + \delta\mathbf{u}) - (f \circ g_t)(\mathbf{x} + \delta\mathbf{u})|]$$

$$\leq \frac{c_1 G_f \sigma_0}{\sqrt{b_{g,0}}},$$

where $c_1$ is some constant. The second equality is due to the linearity of expectations. The last inequality follows from Assumption 3.3 and Lipschitzness of $f$. Consequently, we have

$$\mathbb{E}[(f \circ g)_\delta(\mathbf{x}_1) - (f \circ g)_\delta(\mathbf{x}^*)]$$

$$\leq \frac{\hat{R}^2}{2\eta_0 T_0} + 8\sqrt{2\pi} d G_f^2 G_g^2 \eta_0 + \frac{2c_1 G_f \sigma_0}{\sqrt{b_{g,0}}}$$

$$\leq \frac{12\hat{R} G_f G_g \sqrt{d}}{\sqrt{T_0}} + \frac{2c_1 G_f \sigma_0}{\sqrt{b_{g,0}}}.$$

By Lemma 3.7, we have

$$\mathbb{E}\left[(f \circ g)_\delta(\mathbf{x}_1) - (f \circ g)_\delta(\mathbf{x}^*)\right] \leq \mathbb{E}[(f \circ g)(\mathbf{x}_1) - (f \circ g)(\mathbf{x}^*)] + 2G_f G_g \delta.$$

Consequently, one has

$$\mathbb{E}[(f \circ g)_\delta(\mathbf{x}_1) - (f \circ g)_\delta(\mathbf{x}^*)]$$

$$\leq \frac{12\hat{R} G_f G_g \sqrt{d}}{\sqrt{T_0}} + \frac{2c_1 G_f \sigma_0}{\sqrt{b_{g,0}}} + 2G_f G_g \delta.$$

To obtain $\mathbb{E}[(f \circ g)(\mathbf{x}_1) - (f \circ g)(\mathbf{x}^*)] \leq \rho$, we choose

$$\eta_0 = \frac{\hat{R}}{G_f G_g \sqrt{dT_0}}, \quad T_0 = \Theta\left(\frac{\hat{R}^2 G_f^2 G_g^2 d}{\rho^2}\right), \quad b_{g,0} = \Theta\left(\frac{G_f^2 \sigma_0^2}{\rho^2}\right), \quad \delta = \Theta\left(\frac{\rho}{G_f G_g}\right).$$

$\square$

### D.2 Proof of Corollary 5.3

*Proof.* The total stochastic function oracle calls can be bounded by

$$\mathcal{O}\left(Tb_f b_g + T_0 b_{g,0}\right)$$

$$= \mathcal{O}\left(\frac{G_f G_g \sqrt{d}(\rho + G_f G_g \delta)}{\delta \epsilon^2} \cdot \frac{dG_f^2 G_g^2}{\epsilon^2} \cdot \frac{d^2 G_f^2 \sigma_0^2}{\delta^2 \epsilon^2} + \frac{d\hat{R}^2 G_f^4 G_g^2 \sigma_0^2}{\rho^4}\right)$$

$$= \mathcal{O}\left(\frac{d^{3.5} G_f^5 G_g^3 \sigma_0^2 \rho}{\delta^3 \epsilon^6} + \frac{d^{3.5} G_f^6 G_g^4 \sigma_0^2}{\delta^2 \epsilon^6} + \frac{d\hat{R}^2 G_f^4 G_g^2 \sigma_0^2}{\rho^4}\right)$$

$$= \mathcal{O}\left(\frac{d^3 \hat{R}^{0.4} G_f^{4.8} G_g^{2.8} \sigma_0^2}{\delta^{2.4} \epsilon^{4.8}} + \frac{d^{3.5} G_f^6 G_g^4 \sigma_0^2}{\delta^2 \epsilon^6}\right).$$

$\square$

### D.3 Proof of Corollary 5.4

*Proof.* The total stochastic function oracle calls can be bounded by

$$\mathcal{O}\left(Tb_f' b_g' + Tb_f b_g/m + T_0 b_{g,0}\right)$$

$$= \mathcal{O}\left(\frac{G_f G_g \sqrt{d}(\rho + G_f G_g \delta)}{\delta \epsilon^2} \cdot \frac{dG_f G_g}{\epsilon} \cdot \frac{d^2 \sigma_0^2 G_f^2}{\delta^2 \epsilon^2} + \frac{d\hat{R}^2 G_f^4 G_g^2 \sigma_0^2}{\rho^4}\right)$$

$$= \mathcal{O}\left(\frac{d^{3.5} G_f^4 G_g^2 \sigma_0^2 \rho}{\delta^3 \epsilon^5} + \frac{d^{3.5} G_f^5 G_g^3 \sigma_0^2}{\delta^2 \epsilon^5} + \frac{d\hat{R}^2 G_f^4 G_g^2 \sigma_0^2}{\rho^4}\right)$$

$$= \mathcal{O}\left(\frac{d^3 \hat{R}^{0.4} G_f^4 G_g^2 \sigma_0^2}{\delta^{2.4} \epsilon^4} + \frac{d^{3.5} G_f^5 G_g^3 \sigma_0^2}{\delta^2 \epsilon^5}\right).$$

$\square$

