# OpenReview forum: "Gradient-Free Methods for Nonconvex Nonsmooth Stochastic Compositional Optimization"
_NeurIPS.cc/2024/Conference — NeurIPS 2024 poster_

### Official Review · Reviewer_VEqR · 2024-06-20

**Soundness:** 3
**Presentation:** 3
**Contribution:** 3
**Rating:** 6
**Confidence:** 3

**Summary:**

The authors propose a zero-th order algorithm for Stochastic Compositional Optimization (SCO) problems. Such problems are given in the form of a composition of two functions, each of them being random depending, and the corresponding objective should be minimizer in expectation. The authors consider the nonsmooth nonconvex setting and propose an algorithm which to tackle such problems. The algorithm is called GFCOM, and a variance reduction variant, GFCOM+ is proposed. The authors propose a complexity analysis for the proposed method, with explicit estimates for reaching Goldstein approximate stationary points. They extend their analysis to the convex setting and conclude with numerical experiments and a comparison with the method of Kiefer-Wolfowitz.

I did not read all the details of the proof of the variance reduced algorithm and the convex analysis.

**Strengths:**

The paper is well written, the assumptions are clearly presented. The analysis flows well and looks technically solid.

**Weaknesses:**

The numerical section does not seem to use a specific compositional structure, so it constitutes a weak illustration of the relevance of the method. The paper fail to provide a convincing motivating example related to machine learning.

The value of $c$ should be given to actually implement the algorithm.

**Questions:**

Lemma 3.7 and 3.8, what are the precise references to the paper of Lin et al, which part of the paper do the authors refer to?

Are the random variables $\xi$ and $\zeta$ independent? If yes it should probably be stated.

This sentence is misleading: "hard instances have shown that finding an $\epsilon$-stationary point with respect to the Clarke subdifferential of a Lipschitz function in finite time is computationally intractable". Being $\epsilon$ stationary is not the right notion for nonsmooth optimization: even in the convex setting, the subdifferential does not tend to 0. The hardness result of Kornowski and Shamir is actually different from what is suggested by the sentence and relates to the impossibility of getting close to the critical set.

Table 1: what is WS-GFCOM?

This sentence is misleading: "Consequently, they considered a refined notion of approximate stationary point in terms of the Goldstein $\delta$-subdifferential". Kornowski and Shamir do not use this notion.

Section 6: what is the underlying compositional structure? What are the random variables?

Equality (2) is extremely misleading as a different expression is used in the algorithms and proofs.

**Limitations:**

Section 4.2, what is the value of $c$. Without this the algorithm cannot be implemented.

The convex section is probably of more limited interest than the rest.

The paper should probably contain a discussion, comparing the obtained complexity estimate with the more usual zero-th order stochastic optimization literature. What is the added cost of the stochastic compositional structure?

---

> ### Author Rebuttal · Authors · 2024-08-05
>
> We thank the reviewer VEqR for your insightful and detailed review. Here we would like to address your concern.
>
> **Q1:** What are the precise references to the paper of Lin et al.?
>
> **A1:**  Lemma 3.7 refers to Proposition 2.3 of Lin et al. [30], and Lemma 3.8 refers to Lemma D.1 of Lin et al. [30].
> We will include these details in the revision.
>
> **Q2:** Are the random variables $\mathbf{\xi}$ and $\mathbf{\zeta}$ independent?
>
> **A2:**  Yes, these two random variables are independent. We will explicitly state this point in the revision.
>
> **Q3:** The hardness result of Kornowski and Shamir is different from what is suggested by the sentence.
>
> **A3:** We will rephrase this sentence to avoid confusion in the revision.
> Specifically, Zhang et al. [13] showed that no deterministic algorithms can find an $\epsilon$-stationary point with respect to the Clarke subdifferential in finite time.
> They construct the lower bound with a resisting oracle, so their result only holds for deterministic algorithms that generate a fixed sequence.
> Later, Kornowski and Shamir [14] showed that no randomized algorithms can find an $\epsilon$-near approximately stationary point in finite time with a probability of almost 1.
> It can be inferred from their result that no deterministic or randomized algorithm can find an $\epsilon$-stationary point in finite time since the $\epsilon$-stationary point is a tighter convergence notion than the $\epsilon$-near approximately stationary point.
>
> **Q4:** What is WS-GFCOM?
>
> **A4:** It is the warm-started GFCOM method (lines 218-219), which is presented in Section 5 for the convergence analysis of the convex nonsmooth SCO problem.
> The details of WS-GFCOM are presented in Algorithm 3.
>
> **Q5:** Kornowski and Shamir do not use this notion.
>
> **A5:** Thanks for your careful review. The refined notion in terms of Goldstein $\delta$-subdifferential was proposed by Zhang et al. [13]. We will correct the citation in the revision.
>
> **Q6:** What is the underlying compositional structure? What are the random variables?
>
> **A6:** We provide the details for the compositional formulation of these problems below.
>
> 1. For the portfolio management problem (6), we formulate it as
> $$\min \Phi(x)= \mathbb{E}[F(\mathbb{E}[G(x;\zeta)];\xi)]$$
> where
>  $$G(x;\zeta)= \left[x_1, \ldots, x_N,  \langle r_{\zeta}, x \rangle\right]^{\top}$$
> and
> $$F(w;\xi)=- \langle r_{\xi}, w_{[N]}\rangle + (\langle r_\xi, w_{[N]}\rangle - w_{N+1})^2 + \beta (x).$$
> The random variables in the above formulation are $\xi$ and $\zeta$.
> Both of $\xi$ and $\zeta$ are uniformly sampled from $\\{1,\ldots,T\\}$.
> 2. For the Bellman residual minimization problem (after line 271), we formulate it as
> $$\\min\\Phi(w)=\\mathbb{E}[F(\\mathbb{E}[G(w;\\zeta)];\\xi)]$$
> where
> $$ G(w; \zeta) = [\langle \psi_1, w \rangle,  r_{1, \zeta_1} + \gamma \langle \psi_{\zeta_1}, w \rangle, \ldots, \langle \psi_n, w \rangle,  r_{n, \zeta_n} + \gamma \langle \psi_{\zeta_n}, w \rangle]^{\top}$$
> and
> $$ F(z; \xi) = h(z_{2 \xi} - z_{2\xi+1}).$$
> The random variables in above formulation are $\zeta = [\zeta_1, \ldots, \zeta_n]^{\top}$ and $\xi$. Specifically, each $\zeta_i$ is uniformly sampled from $\\{P_{i 1}, \ldots, P_{i,n}\\}$ and $\xi$ is uniformly sampled from $\\{1,\dots,T\\}$.
>
> We will provide the above explicitly compositional formulation in the revision.
>
> **Q7:** Equality (2) is extremely misleading as a different expression is used in the algorithms and proofs.
>
> **A7:** Thanks for your careful review. We will use different notations in Equality (2) to distinguish $v_t$ in the algorithm in the revision.
>
> **Q8:** What is the value of $c$.
>
> **A8:** According to Lemma 8 of Duchi et al. [33], we can take $c=1$ when $\mathcal{P}$ is the uniform distribution in the unit ball.
> We will provide this explanation in the revision.
>
> **Q9:** The convex section is probably of more limited interest than the rest.
>
> **A9:** We agree that the contribution for the nonconvex case is more interesting, while the convex case is also an important class of problems.
> In Section 5, we show that convexity can result in the improved convergence guarantee to find the $(\delta, \epsilon)$-Goldstein stationary point.
>
> **Q10:** The paper should probably contain a discussion, comparing the obtained complexity estimate with the more usual zero-th order stochastic optimization literature.
>
> **A10:** Thanks for your suggestion. We present the function query complexity of zeroth-order methods for different classes of stochastic optimization problems in the following table.
>
> | Methods  | Problem | Complexity | Reference |
> | -------- | ------- | ------- | ------- |
> | GFM | $\min f(x)$ | $\mathcal{O}(d^{1.5}\delta^{-1}\epsilon^{-4})$ | [30] |
> | GFM+ | $\min f(x)$ | $\mathcal{O}(d^{1.5}\delta^{-1}\epsilon^{-3})$ | [31] |
> | Online to Nonconvex | $\min f(x)$ | $\mathcal{O}(d \delta^{-1}\epsilon^{-3})$ | [32] |
> | GFCOM | $\min f(g(x))$ | $\mathcal{O}(d^{3.5} \delta^{-3}\epsilon^{-6})$ | Corollary 4.2 |
> | GFCOM+ | $\min f(g(x))$ | $\mathcal{O}(d^{3.5} \delta^{-3}\epsilon^{-5})$ | Corollary 4.4 |
>
> **Reference**
> - [13] Jingzhao Zhang, Hongzhou Lin, Stefanie Jegelka, Suvrit Sra, and Ali Jadbabaie. Complexity of finding stationary points of nonconvex nonsmooth functions. ICML 2020.
> - [14] Guy Kornowski, and Ohad Shamir. Oracle complexity in nonsmooth nonconvex optimization. Neurips 2021.
> - [30] Tianyi Lin, Zeyu Zheng, and Michael Jordan. Gradient-free methods for deterministic and stochastic nonsmooth nonconvex optimization. Neurips 2022.
> - [31] Lesi Chen, Jing Xu, and Luo Luo. Faster gradient-free algorithms for nonsmooth nonconvex stochastic optimization. ICML 2023.
> - [32] Guy Kornowski, and Ohad Shamir. An algorithm with optimal dimension-dependence for zero-order nonsmooth nonconvex stochastic optimization. JMLR 2024.
> - [33] John C. Duchi, Peter L. Bartlett, and Martin J. Wainwright. Randomized smoothing for stochastic optimization. SIAM Journal on Optimization 22.2 (2012): 674-701.

---

> > ### Comment · Reviewer_VEqR · 2024-08-08
> > **Answer to rebuttal**
> >
> > I have read the authors response. Thanks.

---

### Official Review · Reviewer_nNfG · 2024-07-10

**Soundness:** 2
**Presentation:** 4
**Contribution:** 2
**Rating:** 5
**Confidence:** 4

**Summary:**

This paper investigates stochastic compositional optimization (SCO) problems, which are popular in many real-world applications. The authors focus on nonconvex and nonsmooth SCO, and propose gradient-free stochastic methods for finding the $(\delta,\epsilon)$-Goldstein stationary points of such problems with non-asymptotic convergence rates. Furthermore, they also use variance reduction technique to accelerate their algorithms with improved results.

**Strengths:**

* The study of the SCO problem is highly significant, and the research on gradient-free algorithms is also of considerable value.
* This paper is well-written and easy to understand.

**Weaknesses:**

* The technical novelty of this work is limited, as the proposed algorithms are mostly based on SGFM [1].
* The authors do not prove that $\mathbf{v}\_t$ is an unbiased gradient estimator of $\nabla \Phi_\delta (\mathbf{x}_t)$ (See Question 1 for details).
* The authors utilize variance reduction technique to accelerate their algorithms. However, they do not review some important related work on variance reduction in this paper, e.g., SVRG, STORM and so on.

[1] Lin et al. Gradient-free methods for deterministic and stochastic nonsmooth nonconvex optimization. NeurIPS, 2022.

**Questions:**

* My primary concern is whether $\mathbf{v}\_t$ in Algorithm 1 is an unbiased estimator of $\nabla \Phi\_\delta (\mathbf{x}\_t)$. Although Lin et al. (2022) have proven that $\mathbf{v}\_t$  in Eq. (2) is an unbiased estimator of $\nabla f\_\delta (\mathbf{x})$, such gradient estimation cannot be directly applied to the SCO problem by introducing auxiliary variables $\mathbf{y}\_t$ and $\mathbf{z}\_t$ in Eq. (3). Specifically, although the estimation of each layer function and its gradients is unbiased, i.e., $\mathbb{E}\_{\xi\_1} [F(\mathbf{x};\xi\_1)]=f(\mathbf{x})$, $\mathbb{E}\_{\xi\_1} [\nabla F(\mathbf{x};\xi\_1)]=\nabla f(\mathbf{x})$, and $\mathbb{E}\_{\xi\_2} [G(\mathbf{x};\xi\_2)]=g(\mathbf{x})$, the main challenge in SCO lies in obtaining an unbiased estimation of gradient $\nabla f(g(\mathbf{x}))$. This is because the expectation over $\xi\_2$ cannot be moved inside of $\nabla f$ such that $\mathbb{E}_{\xi_1,\xi_2} [\nabla F(G(\mathbf{x};\xi_2);\xi_1)]\neq \nabla f(g(\mathbf{x}))$.
* The proposed algorithms in this paper appear to be a direct application of existing algorithms [1] to the SCO problem. Additionally, the accelerated algorithms employ variance reduction technique, which have already been widely used for other  settings in SCO problems [2,3]. Could you elaborate more details on the technical challenge and novelty of your work?

[1] Lin et al. Gradient-free methods for deterministic and stochastic nonsmooth nonconvex optimization. NeurIPS, 2022.

[2] Yuan et al. Efficient smooth non-convex stochastic compositional optimization via stochastic recursive gradient descent. NeurIPS, 2019.

[3] Zhang and Xiao. A stochastic composite gradient method with incremental variance reduction. NeurIPS, 2019.

**Limitations:**

See above

---

> ### Author Rebuttal · Authors · 2024-08-05
>
> We thank the reviewer nNfG for your detailed review.
>
> **Q1:** The authors utilize variance reduction technique to accelerate their algorithms. However, they do not review some important related work on variance reduction in this paper, e.g., SVRG, STORM and so on.
>
> **A1:** Thanks for the suggestion. We will provide a more detailed literature review of variance reduction techniques in the revision, including SVRG [10] and STORM [9].
>
> **Q2:** The authors do not prove that $\\mathbf{v_t}$ is an unbiased gradient estimator of $\nabla \Phi_{\delta}(\mathbf{x}_t)$
>
> **A2:**  We emphasize that our algorithm design and theoretical analysis do not require ${\mathbf v}_t$ to be unbiased.
> In contrast, we focus on providing the upper bound on the expected gradient estimation error, e.g. Lemma B.1 (line 410-415).
> The sharp gradient estimation error indeed leads to reasonable convergence rates (e.g., Theorem 4.1) even if the estimator is biased.
>
> Furthermore, solving the nonconvex stochastic (compositional) optimization problem with a biased gradient estimator is very popular in literature Ref. [2,3,4,5,8,9].
> For example, Zhang and Lin [3] applied the biased gradient estimator to solve a smooth stochastic compositional optimization problem, and their introduction mentioned
> **Using biased gradient estimators can cause various difficulties for constructing and analyzing randomized algorithms, but is often inevitable in dealing with more complex objective functions other than the empirical risk.**
>
> **Q3:**  The proposed algorithms in this paper appear to be a direct application of existing algorithms [1] to the SCO problem. Additionally, the accelerated algorithms employ variance reduction technique, which have already been widely used for other settings in SCO problems [2, 3]. Could you elaborate more details on the technical challenge and novelty of your work?
>
> **A3:** The theoretical results in our work cannot be obtained by combining the techniques of Ref. [1,2,3].
> Notice that the convergence analysis of the existing work [2, 3] heavily depends on the smoothness of both the outer function $f$ (or its differentiable term) and the inner function $g$.
> In this work, both $f$ and $g$ may be nonsmooth,  leading to the nonsmooth compositional function $\Phi(\mathbf{x}) = (f \circ g)(\mathbf{x})$.
> We only apply the randomized smoothing technique [1] on $\Phi(\mathbf{x})$ to construct its smoothing surrogate $\Phi_{\delta}(\mathbf{x}) = (f \circ g)_{\delta}(\mathbf{x})$.
> However, it does NOT lead to any smoothing surrogate of $f$ or $g$, which means we cannot directly combine the analysis of Ref. [1, 2, 3].
> In contrast, we need to use the Lipschitz continuity of $f$ and $g$
> (rather than their smoothness) to carefully bound the expected mean squared error with reasonable sample complexity (see Lemma B.1 and C.1), which is quite different from the analysis in smooth case [1, 2, 3].
>
> **Reference**
> - [1] Tianyi Lin, Zeyu Zheng, and Michael Jordan. Gradient-free methods for deterministic and stochastic nonsmooth nonconvex optimization. Advances in Neural Information Processing Systems 35 (2022): 26160-26175.
> - [2] Huizhuo Yuan, Xiangru Lian, and Ji Liu. Stochastic recursive variance reduction for efficient smooth non-convex compositional optimization. arXiv preprint arXiv:1912.13515 (2019).
> - [3] Junyu Zhang, and Lin Xiao. A stochastic composite gradient method with incremental variance reduction. Advances in Neural Information Processing Systems 32 (2019).
> - [4] Mengdi Wang, Ethan X. Fang, and Han Liu. Stochastic compositional gradient descent: algorithms for minimizing compositions of expected-value functions. Mathematical Programming 161 (2017): 419-449.
> - [5] Mengdi Wang, Ji Liu, and Ethan X. Fang. Accelerating stochastic composition optimization. Journal of Machine Learning Research 18.105 (2017): 1-23.
>  - [6] Yin Liu, and Sam Davanloo Tajbakhsh. Stochastic Composition Optimization of Functions Without Lipschitz Continuous Gradient. Journal of Optimization Theory and Applications 198.1 (2023): 239-289.
> - [7] Quanqi Hu, Dixian Zhu, and Tianbao Yang. Non-smooth weakly-convex finite-sum coupled compositional optimization. Advances in Neural Information Processing Systems 36 (2024).
>  - [8] Cong Fang, Chris Junchi Li, Zhouchen Lin, and Tong Zhang. Spider: Near-optimal non-convex optimization via stochastic path-integrated differential estimator. Advances in neural information processing systems 31 (2018).
>  - [9] Ashok Cutkosky, and Francesco Orabona. Momentum-based variance reduction in non-convex sgd. Advances in neural information processing systems 32 (2019).
>  - [10] Rie Johnson, and Tong Zhang. Accelerating stochastic gradient descent using predictive variance reduction. Advances in neural information processing systems 26 (2013).

---

> > ### Comment · Reviewer_nNfG · 2024-08-09
> >
> > Thank you for your detailed response, which has partially addressed my concerns. For the technical novelty, I have some further questions.
> >
> > **Q:** After reviewing Lemma B.1 and Lemma C.1, I find that the technical contribution in bounding the expected mean squared error may be limited, as the analysis using the Lipschitz continuity seems straightforward and easy. Additionally, I have some concerns regarding the proof of Lemma B.1 (Line 411). Specifically, where does $g_t$ come from in the second inequality? Is this a mistake? If it should be $g$ instead, then the expression in the expectation would be $0$, i.e., $E[\Vert F(g(\mathbf{x}\_t+\delta \mathbf{w}\_{t,j});\xi_{t,j}) - F(g(\mathbf{x}\_t+\delta \mathbf{w}\_{t,j});\xi\_{t,j})\Vert^2]=0$. Is this also a mistake? I think that the authors may omit some steps. Could the authors provide a more detailed and correct analysis of Lemma B.1?

---

> ### Author Response · Authors · 2024-08-09
>
> Thanks for your careful and detailed reply. We would like to address your questions as follows.
>
> **Q1:** I have some concerns regarding the proof of Lemma B.1 (Line 411). Specifically, where does $g_t$ come from in the second inequality? Is this a mistake? If it should be instead, then the expression in the expectation would be 0, i.e., $E[\|F(g(x_t + \delta w_{t,j});\xi_{t,j})-F(g(x_t + \delta w_{t,j}); \xi_{t,j})\|^2]=0$. Is this a mistake?
>
> **A1:** Please note that we have defined $g_t(\cdot)$ for GFCOM (Algorithm 1) at the beginning of Appendix A (line 399), that is
> $$g_t(x) = \frac{1}{b_g} \sum_{i \in [b_g]}G(x; \zeta_{t,i})$$
> which is indeed different from
> $$g(x) = E_\zeta[G(x;\zeta)].$$
> In the proof of Lemma B.1, the notation $g_t(\cdot)$ is **not** a mistake.
> Therefore, the terms
>
> $F(g(x_t + \delta w_{t,j}); \xi_{t,j})$  and   $F(g_t(x_t + \delta w_{t,j}); \xi_{t,j}) $
>
> are **different**.
> The expression $E[\|F(g(x_t + \delta w_{t,j}); \xi_{t,j})-F(g_t(x_t + \delta w_{t,j}); \xi_{t,j})\|^2]$ is also **not** a mistake and cannot be replaced by
> $E[\|F(g(x_t + \delta w_{t,j}); \xi_{t,j})-F(g(x_t + \delta w_{t,j}); \xi_{t,j})\|^2].$
>
> **Q2:** I think that the authors may omit some steps. Could the authors provide a more detailed and correct analysis of Lemma B.1?
>
> **A2:** Recall that we have defined $g_t(x) = \frac{1}{b_g} \sum_{i \in [b_g]}G(x; \zeta_{t, i})$ for the GFCOM method in line 399, we can simplify $y_{t,j}$ and $z_{t,j}$ in Eq.(3) (below line 157) as
>
> $y_{t,j} = g_t(x_t + \delta w_{t,j})$ and  $z_{t,j} = g_t(x_t - \delta w_{t,j}).$
>
> Consequently, we can rewrite $v_t$ (Line 6 in Algorithm 1) as
> $$v_t  = \frac{1}{b_f} \sum_{j \in [b_f]} \frac{d}{2 \delta}(F(g_t(x_t + \delta w_{t,j}); \xi_{t,j}) - F(g_t(x_t - \delta w_{t,j}); \xi_{t,j})) w_{t,j}.$$
>
> Now we can bound the gradient estimation error as
> $$E[\lVert v_t - \nabla \Phi_{\delta} (x_t) \rVert^2] \leq  2 E\Big[ \Big\lVert v_t - \frac{1}{b_f} \sum_{j \in [b_f]} \frac{d}{2 \delta} (F(g(x_t + \delta w_{t,j}); \xi_{t,j}) - F(g(x_t - \delta w_{t,j});\xi_{t,j})) \Big\rVert^2 \Big] + 2 E \Big[ \Big\lVert \frac{1}{b_f} \sum_{j \in [b_f]} \frac{d}{2 \delta} (F(g(x_t + \delta w_{t,j}); \xi_{t,j}) - F(g(x_t - \delta w_{t,j});\xi_{t,j})) - \nabla (f \circ g)_{\delta}(x_t) \Big\rVert^2 \Big] ,$$
> where the inequality follows the fact $\lVert a + b\rVert^2 \leq 2 \lVert a \rVert^2 + 2\lVert b\rVert^2$.
> For the first term on the right-hand side, we have
> $$
> \begin{aligned}
> & 2 E\Big[ \Big\lVert v_t - \frac{1}{b_f} \sum _{j \in [b_f]} \frac{d}{2 \delta} (F(g(x _t + \delta w _{t,j}); \xi _{t,j}) - F(g(x _t - \delta w _{t,j});\xi _{t,j})) \Big\rVert^2 \Big] \\\\
>  \leq & 2 \Big(\frac{d^2}{2 \delta^2 b _f} \sum _{j \in [b _f]} E\Big[\Big\lVert F(g(x _t + \delta w _{t,j}); \xi _{t,j}) - F(g _t(x _t + \delta w _{t,j}); \xi _{t,j})\Big\rVert^2 \Big] + \frac{d^2}{2 \delta^2 b _f} \sum _{j \in [b_f]}E \Big[\Big\lVert F(g(x _t - \delta w _{t,j}); \xi _{t,j}) - F(g _t(x _t - \delta w _{t,j}); \xi _{t,j}\Big)^2\Big] \Big) \\\\
> \leq & 2 \Big(\frac{G _f^2 d^2}{2 \delta^2 b_f} \sum _{j \in [b_f]} E \Big[\Big\lVert g(x _t + \delta w _{t,j}) - g _t(x _t + \delta w _{t,j}) \Big\rVert^2 \Big] + \frac{G _f^2 d^2}{2 \delta^2 b_f} \sum _{j \in [b_f]} E \Big[\Big\lVert g(x _t - \delta w _{t,j}) - g _t(x _t - \delta w _{t,j}) \Big\rVert^2 \Big]\Big)
> \end{aligned}
> $$
> where first inequality is due to $\lVert a + b\rVert^2 \leq 2 \lVert a \rVert^2 + 2\lVert b\rVert^2$ and the second inequality follows Assumption 3.1.
>
> Since $g_t(\cdot)$ is an unbiased estimator of $g(\cdot)$,  Assumption 3.3 implies
>
> $E[\lVert g(x_t + \delta w_{t,j}) - g_t(x_t + \delta w_{t,j}) \rVert^2 ] \leq \frac{\sigma_0^2}{b_g}$ and $E[\lVert g(x_t - \delta w_{t,j}) - g_t(x_t - \delta w_{t,j}) \rVert^2 ] \leq \frac{\sigma_0^2}{b_g}$.
>
> Combining the above two results, we have
> $$2E\Big[ \Big\lVert v_t - \frac{1}{b_f} \sum_{j \in [b_f]} \frac{d}{2 \delta} (F(g(x_t + \delta w_{t,j}); \xi_{t,j}) - F(g(x_t - \delta w_{t,j});\xi_{t,j}))\Big\rVert^2 \Big] \leq \frac{2d^2 G_f^2 \sigma_0^2}{\delta^2 b_g}.$$
> In addition, Lemma 3.8 implies
> $$2 E \Big[ \Big\lVert \frac{1}{b_f} \sum_{j \in [b_f]} \frac{d}{2 \delta} (F(g(x_t + \delta w_{t,j}); \xi_{t,j}) - F(g(x_t - \delta w_{t,j});\xi_{t,j})) - \nabla (f \circ g)_{\delta}(x_t)\Big\rVert^2 \Big] \leq \frac{32 \sqrt{2 \pi} d G_f^2 G_g^2}{b_f}.$$
> Putting everything together, we get the bound
> $$E[\lVert v _t - \nabla \Phi _{\delta} (x _t) \rVert^2 ] \leq \frac{2 d^2 G_f^2 \sigma_0^2}{\delta^2 b _g}  + \frac{32 \sqrt{2 \pi} d G _f^2 G _g^2}{b _f},$$
> which finishes the proof of Lemma B.1.
>
> If you are still confused about our response, you are welcome to discuss it further!

---

> > ### Comment · Reviewer_nNfG · 2024-08-11
> >
> > Thanks for your detailed response! The concerns regarding the correctness of this paper have been fully addressed. The results of this work are indeed valuable. I will increase my score to 5.

---

> > > ### Author Response · Authors · 2024-08-11
> > >
> > > Thank you for raising our score! We are happy to hear that we have addressed your concerns.

---

### Official Review · Reviewer_ZyfL · 2024-07-13

**Soundness:** 3
**Presentation:** 3
**Contribution:** 2
**Rating:** 5
**Confidence:** 3

**Summary:**

This work proposes two zeroth-order methods (including one variance-reduced method) for solving non-convex non-smooth stochastic compositional optimization (SCO). These two methods are further extended to solving convex non-smooth SCO. Theoretical analysis are provided to show the convergence guarantee of all proposed methods.

**Strengths:**

The main contribution of this work is that it proposes the first zeroth-order methods for non-smooth SCO under non-convex and convex setting and presents convergence analysis. The paper is well-written and easy to follow.

**Weaknesses:**

My main concern is in the novelty of the proposed methods and their convergence analysis. Base on my understanding, the proposed four methods are extensions of the existing work [31]. [31] proposed four zeroth-order methods (GFM, GFM+, WS-GFM, WS-GFM+) for solving non-smooth optimization without compositional structure under both non-convex and convex setting. Based on GFM in [31], the GFCOM in this work simply replaces the stochastic function values in the gradient estimators into stochastic function values of the compositional function, and use large batches to ensure the accuracy of the inner function value estimation. The convergence analysis is thus similar to GFM as well.


Reference.

[31] Lesi Chen, Jing Xu, and Luo Luo. Faster gradient-free algorithms for nonsmooth nonconvex stochastic optimization. In International Conference on Machine Learning, pages 5219–5233. PMLR, 2023.

**Questions:**

Please see the weakness section.

**Limitations:**

No significant limitations in this work.

---

> ### Author Rebuttal · Authors · 2024-08-05
>
> We thank the reviewer ZyfL for your insightful review.
>
> **Q1:**  My main concern is in the novelty of the proposed methods and their convergence analysis. Base on my understanding, the proposed four methods are extensions of the existing work [31].
>
> **A1:**  This is the first work that proposes stochastic algorithms to solve the general nonconvex nonsmooth stochastic compositional problem with non-asymptotic convergence rates. Note that both the algorithm design and the convergence analysis of previous work for nonconvex nonsmooth stochastic compositional optimization problems require additional assumptions such as weak convexity [11] and relative smoothness [10].
>
> Our algorithm design and convergence analysis are not simple extensions of existing work [31]. In this work, we apply the randomized smoothing on $\Phi(\mathbf{x})$ to construct its smoothing surrogate $\Phi_{\delta}(\mathbf{x})$.
> However, achieving a sufficient accurate function value of a surrogate compositional problem $\Phi_{\delta}(\cdot)$ to guarantee convergence is more difficult than the counterpart in the problem without compositional structure. Additionally, obtaining a smoothing surrogate $\Phi_{\delta}(\mathbf{x})$ does not result in any smoothing surrogate of $f$ or $g$.
> Therefore, we have to carefully use the Lipschitz continuity of $f$ and $g$ (rather than the smoothness of their surrogate functions) to bound the expected gradient estimation error (see Lemma B.1 and C.1) with reasonable sample complexity,  which is more complicated than previous work for non-compositional problems that can directly use the smoothness of the surrogate function of objective [31].
>
> **Reference**
> - [10] Yin Liu, and Sam Davanloo Tajbakhsh. Stochastic Composition Optimization of Functions Without Lipschitz Continuous Gradient. Journal of Optimization Theory and Applications 198.1 (2023): 239-289.
> - [11] Quanqi Hu, Dixian Zhu, and Tianbao Yang. Non-smooth weakly-convex finite-sum coupled compositional optimization. Advances in Neural Information Processing Systems 36 (2024).
> - [31] Lesi Chen, Jing Xu, and Luo Luo. Faster gradient-free algorithms for nonsmooth nonconvex stochastic optimization. International Conference on Machine Learning. PMLR, 2023.

---

> > ### Comment · Reviewer_ZyfL · 2024-08-12
> >
> > Thank you for the response. I will raise my score to 5.

---

> > > ### Author Response · Authors · 2024-08-12
> > >
> > > We are glad that our rebuttal helped address your concerns. Thank you for raising the score!

---

### Official Review · Reviewer_iZH7 · 2024-07-15

**Soundness:** 3
**Presentation:** 3
**Contribution:** 3
**Rating:** 6
**Confidence:** 3

**Summary:**

This paper studied the zero-order method for computing an approximately stationary point for a Lipschitz function with a composition structure. The main difficulty lies in the function value evaluation. The composition structure involves multiple expectations, requiring multiple rounds of sampling to obtain a satisfactory estimation. The authors also discuss the situation when the objective function is convex. They show that the complexity can be improved under the convexity assumption.

**Strengths:**

Overall, I think this is an okay paper, in the sense that it makes a meaningful contribution to an important setting that was previously unexplored. The technique is intuitive, and the proof is very neat and clean, which is good and should be easy to follow.

**Weaknesses:**

My impression is that the technique is okay but not that surprising. There are two expectations, so two sampling sequences are needed to evaluate the function value. I would say the technical contribution is a little bit insufficient, but I do not strongly oppose it solely on this point.

The following are minor points:
* I would recommend specifying the distribution P in the main text, rather than in the statement of Lemma 3.7. It seems the definition of f_\delta appears in L134, with a very general distribution P following. The definition of this P is not specified even in Algorithms 1 and 2.
* L118: \mathbb{B}(x, \delta) should be \mathbb{B}_\delta(x), according to L93.
* Theorem 4.1, Corollary 4.2, Theorem 4.3, etc.: The upper bound on gradient norm should be in the sense of expectation, right? It seems highly non-trivial to exactly select one R \in [T] such that the gradient norm is minimal in the telescoping sum, due to the difficulty in evaluating the gradient of the smoothed function.
* For Section 5, I recommend considering studying tighter stationarity notions for convex functions, e.g., those whose definition requires weak convexity. The notion of a Goldstein approximate stationary point is rather loose. Thus, whenever possible, considering tighter solution notions would be better.

**Questions:**

See above.

**Limitations:**

See above.

---

> ### Author Rebuttal · Authors · 2024-08-05
>
> We thank Reviewer iZH7 for your careful review and insightful suggestions.
>
> **Q1:**  I would recommend specifying the distribution $P$ in the main text, rather than in the statement of Lemma 3.7. It seems the definition of $f_\delta$ appears in L134, with a very general distribution P following. The definition of this P is not specified even in Algorithms 1 and 2.
>
> **A1:** Thanks for your suggestion. We will explicitly state that $P$ presents the uniform distribution in the unit ball in the revision.
>
> **Q2:** L118: $\mathbb{B}(x, \delta)$ should be $\mathbb{B}_\delta(x)$, according to L93.
>
> **A2:** Thanks for your careful review. We will unify the notations in the revision.
>
> **Q3:** Theorem 4.1, Corollary 4.2, Theorem 4.3, etc.: The upper bound on gradient norm should be in the sense of expectation, right?
>
> **A3:** Thanks for your careful review. It should be the gradient norm in expectation, and we will fix it in the revision.
>
> **Q4:** For Section 5, I recommend considering studying tighter stationarity notions for convex functions, e.g., those whose definition requires weak convexity. The notion of a Goldstein approximate stationary point is rather loose. Thus, whenever possible, considering tighter solution notions would be better.
>
> **A4:** We agree that $(\delta, \epsilon)$-goldstein stationary point is not a tight notion for convex functions.
> For the convex problem, we can study other notions such as the optimal function value gap and the nearly approximate stationary point (Davis et al. [a]).
> -  For the optimal function value gap,  Lemma 5.2 has shown that we can achieve $\mathbb{E}[\Phi(x_T)- \Phi^*]\leq \rho$ with $\mathcal{O}(d \hat{R}^2 G_f^4 G_g^2 \sigma_0^2\rho^{-4})$ function value query calls.
> - For the nearly approximately stationary point, Davis et al. [a] have shown that the explicit convergence rates can be achieved by minimizing the weakly-convex (possibly nonsmooth) function.  We think the convergence analysis based on this notion can also be achieved in the compositional optimization problem by introducing the proximal point iterations in the algorithm design.
>
> **Reference**
> - [a] Damek Davis, and Benjamin Grimmer. Proximally guided stochastic subgradient method for nonsmooth, nonconvex problems. SIAM Journal on Optimization 29.3 (2019): 1908-1930.

---

### Decision · Program_Chairs · 2024-09-25

**Decision:**

Accept (poster)

**Comment:**

The paper considers methods (GFCOM, GFCOM+) to solve the problem of stochastic composite optimization (composition of two functions which are each stochastic) under very relaxed assumptions - notably no smoothness nor convexity assumptions. The methods proposed are zeroth-order in the sense of function iterations and are designed to find an approximate Goldstein stationary point. Analysis is given for both the convex and nonconvex cases and some numerical experiments are included as well.

It is agreed that the paper is well-written and clear in its presentation, i.e., all mathematical statements are precise and theorems are rigorously proven. It is also agreed by all reviewers that the method is interesting.

The weaknesses brought up by reviewers focus on the novelty of the work and the impact of the numerical experiments. Some reviewers called the results unsurprising. However, although the algorithm itself is not particularly innovative it seems that all reviewers agree that the analysis is novel and interesting (after the rebuttal).

Most reviewers engaged in the rebuttal discussion, although some simply opted to keep their score. One reviewer raised their score from a four to a five while another raised their score from a three to a five. It seems that the criticisms the reviewers initially had about the novelty of the work and any technical hangups were sufficiently addressed in the rebuttal stage. The technical soundness of the work has been confirmed by the reviewers’ response to the rebuttal. I did not feel the need to discuss the work with the reviewer after the rebuttal discussion because it was clear from the discussions that had already taken place that the problems were addressed and this was a work with mostly positive reviews.

I recommend to accept the paper.